# Ocean's 8: The Quest for Universal Master Key Filters in DS-CNNs

**Zahra Babaiee**\*
Technische Universität Wien
Vienna, Austria
zahra.babaiee@tuwien.ac.at

**Peyman M. Kiasari**\*
Technische Universität Wien
Vienna, Austria
peyman.kiasari@tuwien.ac.at

**Daniela Rus**
Massachusetts Institute of Technology
Cambridge, MA
rus@mit.edu

**Radu Grosu**
Technische Universität Wien
Vienna, Austria
radu.grosu@tuwien.ac.at

## Abstract

A recent study has proposed the "Master Key Filters Hypothesis" for convolutional neural network filters. This paper extends this hypothesis by radically constraining its scope to a single set of just 8 universal filters that depthwise separable convolutional networks inherently converge to. While conventional DS-CNNs employ thousands of distinct trained filters, our analysis reveals these filters are predominantly linear shifts (ax+b) of our discovered universal set. Through systematic unsupervised search, we extracted these fundamental patterns across different architectures and datasets. Remarkably, networks initialized with these 8 unique frozen filters achieve over 80% ImageNet accuracy, and even outperform models with thousands of trainable parameters when applied to smaller datasets. The identified master key filters closely match Difference of Gaussians (DoGs), Gaussians, and their derivatives, structures that are not only fundamental to classical image processing but also strikingly similar to receptive fields in mammalian visual systems. Our findings provide compelling evidence that depthwise convolutional layers naturally gravitate toward this fundamental set of spatial operators regardless of task or architecture. This work offers new insights for understanding generalization and transfer learning through the universal language of these master key filters. Code is available at: https://github.com/ranaa-b/MasterKeyFilters

## 1 Introduction

Convolutional Neural Networks (CNNs) have significantly advanced computer vision through their hierarchical representations using trainable filters. As architectures evolved toward greater performance, models such as VGG [28], ResNet [10], and DenseNet [14] incorporated thousands of filters across their layers. This trend continued with the development of Depthwise Separable Convolutional Neural Networks (DS-CNNs) [13, 12], which separate spatial and channel-wise computations for improved efficiency. Contemporary architectures like the ConvNeXt family [24, 34] utilize DS-CNNs with up to 50,000 trainable spatial filters.

Recent research identified a notable pattern in trained depthwise convolutional kernels across various DS-CNN architectures [2, 4]. Through analysis of trained filters using unsupervised clustering, they demonstrated that these patterns converge into distinct clusters resembling Difference of Gaussian

---

\*Equal contribution.

39th Conference on Neural Information Processing Systems (NeurIPS 2025).

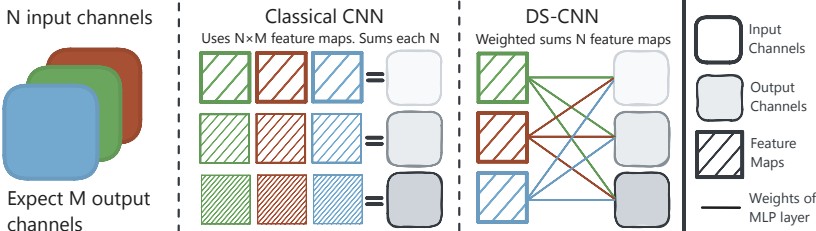

Figure 1: **Comparison of Classical CNN and DS-CNN architectures:** Left: Input with N channels. Center: In Classical CNNs, each output channel is produced by convolving a unique filter with each input channel, followed by summing the resulting feature maps. This results in N×M distinct filters and corresponding feature maps. Right: DS-CNN uses only N filters (one per input channel) to create N feature maps, then applies an N×M MLP layer to linearly combine these feature maps into M output channels. DS-CNNs represent a parameter-efficient *subset* of classical CNNs, reducing the number of required convolutional operations.

(DoG) functions and their derivatives. Their study classified over 95% and 90% of filters from ConvNextV2 and ConvNeXt models, respectively, into Gaussian-related clusters, indicating consistent patterns in filter learning.

Subsequently, another work proposed the "Master Key Filters Hypothesis," [5] proposing that there exist master key filter sets that are general for visual data, and that the depthwise filters in DS-CNNs tend to converge to these master key filters, regardless of the specific dataset, task, or architecture. This hypothesis challenges the conventional understanding that convolutional filters become increasingly specialized in deeper layers and suggests instead that a set of fundamental filters may underlie the performance of these networks. In this paper, we extend the "Master Key Filters Hypothesis" by radically constraining its scope through identification of a minimal fundamental master key filter set. While the original hypothesis posited the existence of general-purpose filter sets for visual data—potentially comprising numerous filters across multiple sets—our systematic unsupervised analysis across architectures and datasets reveals a remarkably compact representation. We demonstrate that DS-CNNs predominantly converge toward a basis of just 8 distinct filters, where a substantial proportion of learned filters approximate linear shifts of these fundamental kernels. Notably, networks restricted to this compact basis maintain performance integrity, suggesting these filters capture essential visual processing primitives rather than task-specific features. This finding significantly refines the original hypothesis by establishing both the cardinality and specific form of a universal filter basis for visual computing.

**Linear Shift:** Tensors $X$ and $Y$ are linear shifts iff $X = aY + b$ where $a$ and $b$ are scalars.

Our work also substantially refines the observations made in [2], which identified Gaussian-related patterns in DS-CNN filters without constraining their potential variability. While that study demonstrated the prevalence of Guassian-like filters, it allowed for an effectively infinite continuum of these structures with arbitrary standard deviations and noise characteristics. In contrast, we're narrowing it down as linear shift of a mere 8 fundamental filters. This significantly narrows the theoretical space.

These identified filters correspond to mathematical forms matching Difference of Gaussians (DoGs), Gaussians, and their derivatives, which are established components in scale-space theory [20] and share structural similarities with receptive fields observed in mammalian visual systems [35, 36]. Networks initialized with these 8 filters achieve over 80% ImageNet accuracy and demonstrate superior performance compared to models with thousands of trainable parameters when applied to smaller datasets.

Our findings provide empirical support for the "Master Key Filters Hypothesis" and suggest potential applications in efficient network design and transfer learning, while contributing to the understanding of generalizable patterns in visual processing systems.

## 2 Related Work

**Depthwise Convolutional Filters.** Depthwise Convolutions (DCs) have revolutionized the design of Convolutional Neural Networks (CNNs) by using only one feature map per input channel, leading to the development of lightweight and performant architectures such as MobileNet [13], Efficient-Net [30], and ConvNeXt [24]. As illustrated in Figure 1, classical CNNs utilize $c_{in} \times c_{out}$ completely separate filters, creating independent feature maps for each input-output channel combination. Each output channel is computed as:

$$Y_i = \sum_{j=1}^{c_{in}} K_{i,j} * X_j, \text{for } (i,j) \in [c_{out}] \times [c_{in}]$$

where $X \in \mathbb{R}^{H \times W \times c_{in}}$ is the input tensor with $c_{in}$ channels, $Y_i$ is the $i$-th output channel, $K_{i,j}$ is the convolutional kernel for the $i$-th output channel and $j$-th input channel, and $[n]$ denotes the set $\{1, 2, ..., n\}$.

In contrast, DS-CNNs force the model to use only $c_{in}$ feature maps (one per input channel) followed by linearly combining these feature maps using pointwise convolutions or MLP layers. It is important to emphasize that DS-CNNs combine the feature maps resulting from the depthwise convolutions rather than linearly combining the kernels directly:

$$Y_i = \sum_{j=1}^{c_{in}} W_{i,j} \cdot K_j * X_j, \text{for } (i,j) \in [c_{out}] \times [c_{in}]$$

Although DS-CNNs are restricted-CNNs, but this restriction significantly reduces the parameter count while maintaining competitive performance. They have demonstrated superior efficiency and accuracy within their class size on diverse tasks-including ImageNet classification, scene understanding, and graph conceptualization-compared to traditional CNNs and vision transformers[24, 34, 3].

Recent studies have revealed striking properties of depthwise convolutional filters in these networks. Trockman et al. [33] observed that learned filters in their DS-CNN model ConvMixer exhibit highly structured covariance matrices. Furthermore, Babaiee et al. [2] discovered that trained depthwise convolutional kernels across all layers of DS-CNNs converge into a few main clusters, each resembling the difference of Gaussian (DoG) functions and their first and second-order derivatives. The authors were able to classify the majority of the filters from state-of-the-art DS-CNN models. Building on this work, recent work. [5] introduced the "Master Key Filters Hypothesis," which proposes that depthwise filters in DS-CNNs exhibit generality across domains, architectures, and layer depths, challenging the conventional view that deeper layers become increasingly specialized. Our work builds upon these findings by investigating the potential of using a limited number of unique filters in DS-CNNs, exploiting the observed clusterability in depthwise convolutional kernels, and moving towards finding the master key filters.

**Filter Diversity**. Filter pruning and compression techniques have been widely explored to reduce the computational complexity and memory footprint of CNNs [11]. Structured pruning in CNNs is typically achieved by removing redundant filters [19]. These methods highlight the importance of "feature-map" diversity in CNNs, as removing redundant or less informative filters can lead to more efficient models without significant performance degradation. In contrast, our work is not attempting to reduce the number of parameters or computational cost, but rather investigating the role of "filter" diversity in DS-CNNs and challenging the assumption that a large number of unique filters is necessary for optimal performance. By discovering that a small set of carefully chosen filters can effectively replace a large number of learned filters in DS-CNNs, we are shedding light on the inherent limited diversity present in the learned depthwise filters.

**Scale-Space Theory.** Scale-space theory, which examines signals across different scales, was initially developed in the mid-1980s [15] and has since become a fundamental concept in signal processing, particularly within the field of computer vision. Lindeberg [20] introduced a computational framework for visual receptive fields that exploits symmetry properties across space and time. This framework is compelling for two reasons [23]: First, it offers a normative perspective on visual processing that closely aligns with the hierarchical stages observed in the visual systems of higher mammals[21, 1]. Second, it provides a provable approach to capture the natural transformations of images over space and time [22]. Gaussian derivatives are the sole kernels satisfying isotropy (rotational invariance)

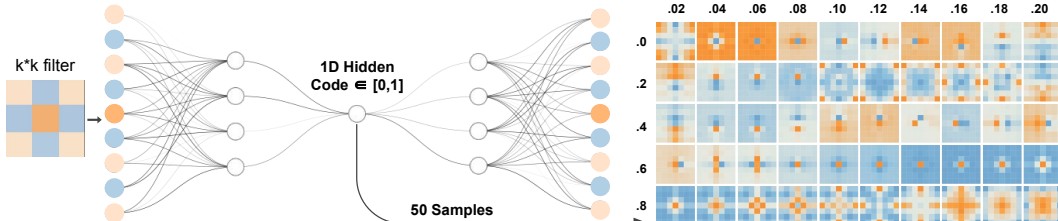

Figure 2: **Visualization of selecting candidate filters using autoencoder-based dimensionality reduction**. Left: An autoencoder compresses filters into a 1D hidden code. Right: Heatmap of 50 uniformly sampled candidate filters from 1D hidden code, generated by the decoder part of the autoencoder. These samples serve as the initial pool for our search for the master key filters.

and non-creativity (with respect to the causality principle) in scale-space theory [20]. Remarkably, our work reveals that the master key filters required for efficient performance in DS-CNNs consists of only 8 distinct filters: Gaussians, Difference of Gaussians (DoG, which can be approximated by the Laplacian of Gaussians), and first derivatives of Gaussians. This finding establishes a strong connection between the principles of scale-space theory and the design of CNN architectures.

## 3 Do We Need Thousands of Distinct Filters?

In this section, we investigate whether employing thousands of unique filters is essential for maintaining the performance of DS-CNNs. In particular, we explore what is the impact on the performance of the network, when we replace the trained filters with a minimal set of distinct filter variations.

### 3.1 The Quest for Master Key Filters

#### 3.1.1 Autoencoder Design

To reduce the number of distinct filters in DS-CNNs, we distilled filters from publicly available trained models of various sizes. Specifically, we gathered all depthwise $7 \times 7$ filters from every layer in our model bank, centered each filter and scaled its L2 norm to 1, and used this normalized collection to train an autoencoder that encodes each filter into a single dimension (Fig. 2), following [2]. This procedure captures the essential filter characteristics while markedly reducing dimensionality.

The autoencoder architecture comprises two primary components: an encoder and a decoder. The encoder consists of four intermediate layers, each followed by a leaky rectified linear unit (Leaky ReLU) activation function. These layers progressively compress and abstract the input filter representations. The final layer of the encoder, known as the code layer, employs a sigmoid activation function to map the compressed filter representations to values within the range [0, 1]. This mapping ensures that the encoded filters are 1 dimensional, thus easy to sample.

The decoder on the other hand, is responsible for reconstructing the original normalized filters from the encoded representations. It mirrors the structure of the encoder, with four intermediate layers that gradually upsample and expand the encoded features. The final layer of the decoder utilizes a hyperbolic tangent (tanh) activation function, which allows for accurate reconstruction of the normalized filters within [-1, 1]. This choice of activation function ensures that reconstructed filters maintain their centering and scale, aligning with the characteristics of the original normalized filters.

After training the autoencoder, we performed uniform sampling from the code layer. We took various numbers of samples, 50, 25, and 10, from the [0,1] interval to generate filter sets (Fig. 2). Using the decoder, we transformed these codes back into filters. We name them "candidate filters".

#### 3.1.2 Linear shift (ax+b) approximation

In a layer, $c$-th channels depthwise filter can be denoted as $\mathbf{F}_c \in \mathbb{R}^{k \times k}$, where $c$ is the index of the channel. When flattened, $F_c$ can be represented as a vector $\mathbf{f}_c \in \mathbb{R}^{k^2}$. The matrix $F$ composed of all flattened vectors of a layer is $F = [\mathbf{f}_1, \mathbf{f}_2, \ldots, \mathbf{f}_C]^T \in \mathbb{R}^{C \times k^2}$.

For each depthwise filter $F_c$ learned by the ConvNeXtV2-tiny model, we identified the scalar coefficients $a$ and $b$, which minimized the Euclidean distance between the corresponding flattened filter vector $f_c$, and the linear combination $af'_k + b$, where $f'_k$ represents a flattened candidate filter. The original filter was then substituted with the optimal linear combination $af'_{k_{min}} + b$ that exhibited the smallest distance to the original.

To solve for scalars $a$ and $b$ that minimize the distance between vectors $f_c$ and $af'_k + b$, we use linear regression. Here, the goal is to determine the coefficients $a$ and $b$ for two vectors $x$ and $y$ such that by having $\tilde{y} = ax + b$ the length of the vector $y - \tilde{y}$ is minimized. This problem has a well-known solution. If the length of our vectors is $l$ we have:

$$a = \frac{l \sum_{i=1}^{l} x_i y_i - \sum_{i=1}^{l} x_i \sum_{i=1}^{l} y_i}{l \sum_{i=1}^{l} x_i^2 - (\sum_{i=1}^{l} x_i)^2} \quad b = \frac{\sum_{i=1}^{l} y_i \sum_{i=1}^{l} x_i^2 - \sum_{i=1}^{l} x_i \sum_{i=1}^{l} x_i y_i}{n \sum_{i=1}^{l} x_i^2 - (\sum_{i=1}^{l} x_i)^2} \quad (1)$$

Calculating Equations (1) can be computationally intensive, especially when dealing with hundreds of thousands of filters. To reduce computational complexity, we can use a normalization trick. Since any linear shift of $x$ does not alter the optimal $\tilde{y}$, we normalize $x$ using the transformation $\hat{x} = \frac{x - \bar{x}}{||x - \bar{x}||}$. With this normalization, $\sum_{i=1}^{l} \hat{x}_i = 0$ and $\sum_{i=1}^{l} \hat{x}_i^2 = 1$, allowing us to simplify Equation(1).

$$a = \frac{l \sum_{i=1}^{l} x_i y_i}{n} = \langle x, y \rangle \quad b = \frac{\sum_{i=1}^{l} y_i}{n} = \bar{y} \quad (2)$$

Consequentially, Given the vectors $\hat{x}_1, \hat{x}_2, \ldots, \hat{x}_n$ as the rows of matrix $\hat{X}$ and the vectors $y_1, y_2, \ldots, y_m$ as the columns of matrix $Y$, we introduce the vector $y_{\text{mean}}$, which contains the means $\bar{y}_1, \bar{y}_2, \ldots, \bar{y}_m$. Using these, we can calculate the coefficients $a_{ij}$ and $b_{ij}$ for each pair of $x_i$ and $y_j$ through matrix multiplication.

$$A = \hat{X}Y \quad B = y_{\text{mean}}\mathbf{1}^\top \quad (3)$$

For each layer, with the set of depthwise filter vectors matrix $F$ and the sample filter vectors matrix $F'$, we calculate the coefficients as above to find the closest linear-shift approximation. We chose linear shifts for approximating the original filters because they preserve the heatmap and essential characteristics of the filters. By applying a linear shift to a filter sample, we maintain the spatial structure and relative importance of different regions within the filter.

### 3.1.3 Evaluation of $F'$

In order to evaluate the impact of replacing the original filters with their linear-shift approximations derived from the sampled filter set, we assessed the performance of the modified models on the test set. In Table 1 we present the accuracy of models with varying sizes from the ConvNeXtv2 and Hornet [27] families, along with their accuracy after their filter replacement. Quite remarkably, when replacing all the filters of the models with approximations based on only 50 sampled filters, the model performance remains robust, even without any fine-tuning. This resilience is particularly evident for larger model sizes. In the case of ConvNeXtv2 Huge, replacing nearly 50K filters with just 50 sampled filters results in less than a two percent accuracy drop, without any fine-tuning.

As expected, reducing the number of sampled filters leads to a larger accuracy gap, and the ConvNeXtv2 models struggle to perform well when using an small set of only 10 filters. However, it is important to note that the filter samples used in this experiment were obtained through uniform sampling from the code layer of the autoencoder. This immediately raises the following question: *Is there a more strategically selected set of filter samples which can yield a better performance?*

To elucidate this question, we focused on the ConvNeXt-v2-Tiny model and conducted a greedy search on a set of 50 filter samples. We began with the 50 uniformly sampled filters and iteratively removed filters one by one. Figure 2 illustrates the 50 filter samples used in this search. At each iteration, we evaluated the model accuracy after removing each individual filter and eliminated the one whose removal resulted in the smallest accuracy drop. This process was repeated for all filters.

Table 1: **Performance comparison when thousands of trained filters are replaced with linear shifts (ax+b) from candidate filters.** *Without **any** fine-tuning* (For trained model see Table 2), models with just 8 selected filters from our greedy search on *ConvNeXtv2 Tiny* maintain remarkably high accuracy (e.g., only 3.5% drop for ConvNeXtv2 Huge despite reducing from 50k to 8 unique filter patterns) and even on different architecture, HorNet. Concidering models' high sensitivity to filter alterations, this is evidence that DS-CNN filters predominantly converge to these filters.

| ConvNeXtv2 Models | ConvNeXt | | | | | Hornet | |
|---|---|---|---|---|---|---|---|
| | **Pico** | **Tiny** | **Base** | **Large** | **Huge** | **Tiny** | **Small** |
| Number of Filters | 2 944 | 6 624 | 18 048 | 27 072 | 49 632 | 11 488 | 17 232 |
| Original Acc | 80.3% | 83.0% | 84.9% | 85.8% | 86.3% | 82.3% | 83.5% |
| Acc with 50 candidates | 75.0% | 75.4% | 80.5% | 83.2% | 84.0% | 79.4% | 81.3% |
| Acc with 25 candidates | 72.0% | 66.9% | 72.8% | 79.6% | 80.4% | 78.3% | 80.9% |
| Acc with 10 candidates | 23.4% | 1.0% | 1.4% | 3.0% | 2.0% | 66.3% | 70.5% |
| Acc with 8 (greedy search) | 73.1% | 76.7% | 79.3% | 81.2% | 82.8% | 76.0% | 78.1% |
| Acc with 8 random filters | 0.11% | 0.10% | 0.10% | 0.12% | 0.09% | 0.96% | 1.0% |

The accuracy plot during the greedy search, as shown in Figure 3, reveals an interesting trend. The model's accuracy remains relatively stable until the last 10 filters are removed, with the curve exhibiting a distinct elbow around 8 samples. This observation suggests that a small subset of only 10 filters is playing a crucial role in maintaining the performance of the model. To further refine the search, we selected the 10 best-performing filters from the previous step and expanded our search space by sampling 4 additional filters around each of these 10 filters. This local exploration allows us to fine-tune the selection of filters and capture any potential variations that may enhance performance.

We then conducted a second round of the greedy search using this expanded set of filters. The search converged to a set of 8 filters located just after the curve elbow. These 8 filters, as shown in Figure 4, represent a highly informative subset that can effectively replace the original large set of filters, while minimally impacting the model's accuracy.

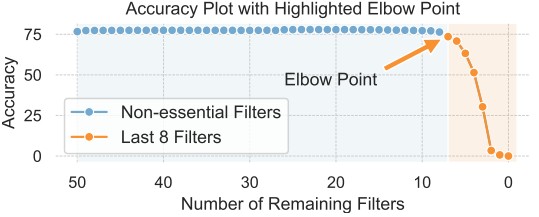

Figure 3: **Our systematic greedy search for the essential filters A.1**. While the removal of most filters did not noticeably change accuracy, 8 of them were essential, consistently in all models we tested.

In order to make sure that our search is not violating the "Test Set Violation" principle, we repeated the experiment as follows. We separated 100 random samples from each of the classes of the ImageNet *training set*. We then used this new set as our evaluation set for the greedy search on the 50 filter samples. We followed the same steps as outlined before, and the search resulted the same 8 filters.

The last row of Table 1 showcases the accuracy of the models, when their filters are replaced by the 8 filter transformations, obtained from the greedy search. Remarkably, the results demonstrate that the ConvNeXtv2 models accuracy achieved with these 8 filters, surpasses even the performance of the 25-filters-sample set. This finding underscores the effectiveness of the greedy search approach in identifying a highly discriminative subset of filters. Moreover, it highlights the potential for replacing a large number of filters, up to 50K in the case of ConvNeXt v2 models, with just 8 strategically selected filters, while maintaining an acceptable performance.

To validate the generalizability of these 8 filter samples, we extended our experiments to the ConvNeXt-v2-Pico model, which represents a different model size. In this case also, we arrived at a similar set of 8 filter samples, indicating the robustness and transferability of our findings across different model architectures. The consistency of the 8 filter samples across different model sizes suggests that these filters capture fundamental and generalizable patterns in the data. It hints at the existence of a set of universal filters that can effectively represent the essential information required for accurate classification.

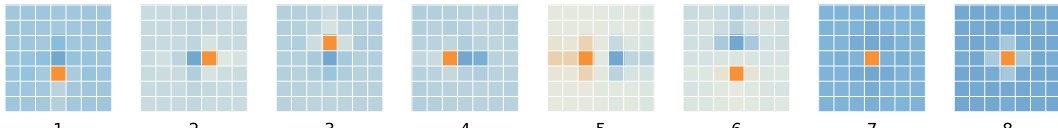

Figure 4: **Heatmap visualization of the eight universal filters discovered through systematic greedy search on the ConvNeXtv2 tiny model.** Our empirical analysis demonstrates that DS-CNN filters predominantly converge to linear shifts (ax+b) of one of these eight filters, regardless of architecture or dataset. Filters 1-4 display central difference operator characteristics, and filters 5-8 correspond to established mathematical image processing fundamentals (See Figure 5).

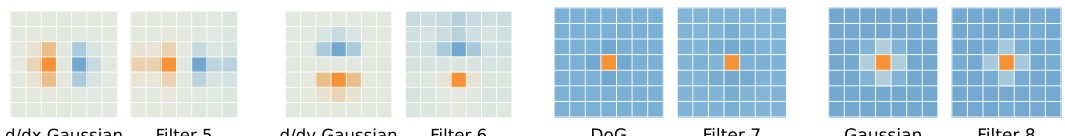

Figure 5: **Correspondence between the empirically discovered filters (5-8) and their theoretical mathematical counterparts.** Left column shows the idealized mathematical forms: first derivatives of Gaussians in x and y directions (filters 5-6), Difference of Gaussians (filter 7), and Gaussian function (filter 8). Right column shows our discovered filters that closely approximate these operators, demonstrating the network's natural convergence toward established visual processing primitives.

The remarkable performance maintained when replacing thousands of trained filters with just our 8 master key filters cannot be coincidental. Noting models' high sensitivity to filter alterations (evidenced by the catastrophic performance drop with 8 random filters), strongly indicates that DS-CNN filters predominantly converge to fundamental patterns during training. Even more compelling, these 8 filters (discovered exclusively from ConvNextV2 Tiny) transfer seamlessly to architecturally distinct models like Hornet. This fascinating cross-architecture generalization suggests that DS-CNNs naturally gravitate toward a mathematically well-defined universal filter code that captures fundamental visual processing operations.

## 3.2 Understanding the Eight Filters

This subsection investigates into the functional characteristics of the eight filters identified through our systematic greedy search, and which have proven to be very effective across various datasets. By analyzing these filters, we aim to understand their resemblance to traditional image-processing operators and their potential roles in effective feature extraction within the network.

**Filters 1-4:** These 4 filters in Figure exhibit characteristics reminiscent of central difference operators, commonly used for approximating Gaussian derivatives discretely. The arrangement and weights of these filters mimic the theoretical models used in edge detection and texture analysis.

**Filters 5-6:** These 2 filters strongly resemble 1st order Gaussian derivatives along the $x$ and $y$ axis. These filter contribute more pronounced spatial smoothing than previous filters. This characteristic enables these filters to capture broader and more varied textural information from the input images, potentially allowing for a better generalization across different visual contexts.

**Filters 7:** This filter resemble a 2-D discrete analogue of the Difference of Gaussians (DoG) due to its positive center with slightly negative surround. The DoG filter, often approximated by the Laplacian of the Gaussian in digital image processing, is crucial for blob detection and bar pattern recognition in images. These filter likely contribute to the model's ability to differentiate areas of rapid intensity change, enhancing edge and contour detection.

**Filter 8:** This filter closely aligns with a very fine-scaled Gaussian kernel. In image processing, Gaussian kernels are smoothing filters used to reduce noise and detail. This results in a blurred image that preserves edges better than uniform filters. Gaussian filters are mathematically proven to be the only function for scale-space representation.

**Filters formal definition:** For completeness, we provide below the formal definition of the continuous functions corresponding to the 2D Gaussian, the 2D derivative of the Gaussian along the $x$ and the $y$

Table 2: **ImageNet Top-1 accuracy comparison between conventional trainings and our 8-filter constraint.** Models restricted to using only our 8 unique filters (plus learnable bias terms) achieve comparable accuracy to their fully-trained counterparts. The consistent performance across different architectures (ConvNeXtv2 and Hornet) demonstrates the universality of these fundamental filters.

| Models | ConvNeXtv2 | | | | Hornet |
| | Pico | Tiny | Base | Large | Tiny |
|---|---|---|---|---|---|
| Number of Original Filters | 2 944 | 6 624 | 18 048 | 27 072 | 11 488 |
| Original model with FCMAE[1] | 80.3% | 82.9%[2] | 84.9% | 85.8% | — |
| Original model | 79.7% | 82.5% | 84.3% | 84.5% | 82.3% |
| Our 8 unique filters + bias | 80.2% | 82.7% | 84.6% | 85.4% | 81.8% |

[1] FCMAE (Fully Convolutional Masked Autoencoder Framework) is a heavy pretraining.
[2] This accuracy is from the model released by the ConvNeXtV2's official paper repository. We trained the model using their exact same script and training parameters (with the default random seed set to 0) and achieved an accuracy of 82.7%, matching the result obtained with our 8-filter configuration.

axis, respectively, and the 2D difference of Gaussians (DoG, Laplacian, Mexican hat):

$$\text{Gaussian:} \quad G(x,y) \quad = \quad e^{-(x^2+y^2)/2\sigma^2} \qquad \Delta\text{Gaussian:} \quad \text{DoG}(x,y) = G_1(x,y) - G_2(x,y)$$

These formulations were used to construct the last four filters as depicted in Figure 5, with the exception of the DoG, for which its approximation, the Ricker wavelet, was used for simplicity. The reconstructed filters bear a strong resemblance to those discovered through our encoding and greedy search methods, validating our hypothesis. in function approximation and emphasizing the practical relevance of traditional image processing theories in modern deep learning architectures.

**Functional Approximation and Construction:** We reconstructed the lower four filters using theoretical formulas typically associated with these image processing techniques, depicted in Figure 5. The reconstructed filters bear a strong resemblance to those discovered through our encoding and greedy search methods, validating our approach in filter selection and emphasizing the practical relevance of traditional image processing theories in modern deep learning architectures.

## 4 Experiments

So far, we've identified 8 unique filters that, when used as linear-shift approximations to replace the filters of trained models, maintain relatively stable accuracy despite this dramatic change in model parameters. These findings naturally lead to a key question: *Can models be successfully trained from initialization with just these 8 filter types kept frozen throughout training?* In this section, we present experimental evaluations on ImageNet and additional datasets to investigate this question.

### 4.1 ImageNet

The results in Table 1 demonstrate that model accuracy remains stable despite significantly reduced filter diversity. In these experiments, model filters are linear shifts of one of the 8 identified filters, mathematically expressed as $a(x + b)$. Given the architecture of DS-CNNs, the coefficient $a$ can be transferred to the fully-connected layers following depthwise convolutions (the following pointwise layer), effectively simplifying the filters to $x + b$, where $b$ acts as a learnable bias. This insight motivated us to train models from scratch using only these 8 fixed filters with learnable biases.

**Training with Only 8 Frozen Filters**

To investigate the effectiveness of our 8 candidate filters, we trained ConvNeXtv2 models from scratch, initializing each layer's filters with these 8 filters while allowing only the bias terms to be trainable. We followed the same 300-epoch training pipeline described in the original paper [34], with the critical difference that all convolutional filters remained frozen throughout training. Table 2 presents our results. Remarkably, the ConvNeXtv2 Tiny model with only 8 types of filters achieved

an accuracy of 82.7%, merely 0.2% lower than the model trained with 6,624 trainable filters and FCMAE pretraining. Similarly, the smaller ConvNeXtv2 Pico model with 8 types of frozen filters reached 80.2% accuracy, just 0.1% below the model with 2,944 trainable filters.

To validate the generalizability of our findings across different architectures, we conducted an additional experiment with the Hornet model [27]—a DS-CNN with substantially different structure than the ConvNext family. The Hornet Tiny model with only our 8 filters achieved 81.8% accuracy compared to 82.3% for the original model, representing only a 0.5% drop. Notably, these 8 filters were derived exclusively from ConvNext models through greedy search on the ConvNeXtv2 Tiny model, yet transferred effectively to Hornet without modification.

It is worth emphasizing that despite our significant architectural modification, these experiments used the original training hyperparameters for each model. A dedicated hyperparameter search optimized for this fixed-filter approach could potentially enhance results further.

Table 3: **Cross-dataset evaluation** demonstrating the superiority of our universal filter approach on smaller datasets. The ConvNeXt Femto model restricted to using only 8 unique frozen filters outperforms both models trained from scratch and those using transferred ImageNet filters, on Oxford Pets and Oxford Flowers. We evaluate multiple model sizes (Atto, Femto, Pico, Tiny) for Flowers and Pets datasets to verify that our observed advantage is consistent across architectures of varying capacity and not merely an artifact of dataset size limitations.

| Dataset | CIFAR10 | STL-10 | Oxford Flowers | | | | Oxford Pets | | | |
|---|---|---|---|---|---|---|---|---|---|---|
| # Training Set Size | 50000 | 5000 | 2040 | | | | 3680 | | | |
| Model / Filters | Femto | Femto | Atto | Femto | Pico | Tiny | Atto | Femto | Pico | Tiny |
| Original (normal traning) | 96.9 | 80.4 | 63.3 | 66.0 | 60.2 | 75.7 | 38.4 | 36.3 | 40.1 | 65.4 |
| ImageNet Transferring | **97.1** | **83.2** | 72.2 | 73.2 | 74.8 | 81.8 | 58.5 | 56.0 | 66.3 | 80.1 |
| Our 8 Unique Filters | 96.3 | 83.1 | **77.8** | **77.7** | **77.2** | **85.1** | **66.5** | **66.4** | **72.8** | **81.8** |

## 4.2 Other Datasets

To investigate the generalizability of our findings, we extend our experiments to other datasets and compare the performance of the ConvNeXt Femto across various settings.

**Datasets and Settings.** We evaluate the low filter variety on four datasets: CIFAR-10 [16], Flowers [25], Pets [26], and STL-10 [8]. These datasets have smaller scales compared to ImageNet, with the size of training sets ranging from 2040 to 50000 samples. We use the ConvNeXt Femto model as our base architecture for all datasets, and additionally use ConvNeXt Atto, Pico, and Tiny for the Flowers and Pets datasets. For a fair comparison, we train the model on all datasets for 300 epochs, following the training parameters from the ConvNeXt paper [24], and keep the training settings consistent across all runs and datasets. For each dataset, we first train the model to obtain baseline accuracy. We then evaluate two filter initialization strategies: (1) depthwise filters transferred from a ConvNeXt model pretrained on ImageNet, and (2) eight frozen filter types trained from scratch. Table 3 shows the resulting accuracies for each setting.

**Results.** The results demonstrate the effectiveness of using the 8 frozen filters across different datasets. Notably, the performance improvement becomes more pronounced as the dataset size decreases. For the Flowers and Pets datasets, the frozen 8 filter types achieve remarkable improvements of up to 11% and 34.5%, respectively, compared to the baseline model. Interestingly, on these smaller datasets, the eight frozen filter types even outperform the transferred filters from the model trained on ImageNet. To further evaluate and verify the performance of our 8 filters on these datasets, we used other sizes of the ConvNeXt model, the results of which showed consistent superior performance on all sizes.

This observation suggests that the carefully selected filter types capture fundamental patterns that are highly relevant to the task at hand, even when the dataset size is limited. This finding has significant implications for scenarios where training data is scarce or computational resources are limited.

# 5   Conclusions

This paper extends the "Master Key Filters Hypothesis" by identifying a set of just 8 filters. While conventional DS-CNNs employ thousands of distinct trained filters, our analysis reveals these filters predominantly converge to linear shifts (ax+b) of one of the filters in our discovered set. This finding significantly narrows the theoretical space proposed in previous work. The discovered filters closely match established mathematical forms: Difference of Gaussians, Gaussians, and their derivatives, creating a bridge between classical computer vision theory and modern deep learning practice. This correspondence to structures found in both scale-space theory and mammalian visual systems suggests DS-CNNs inherently rediscover optimal operators aligned with natural image statistics.

Our systematic experiments demonstrate that networks initialized with these 8 frozen filters achieve over 80% ImageNet accuracy. Particularly noteworthy is the superior performance of our filters on smaller datasets, where models initialized with our filters outperform even ImageNet transfer learning. This suggests that these filters encode fundamental visual processing primitives that transcend specific datasets and visual domains, offering a novel approach to transfer learning.

**Future Work** may explore direct optimization approaches to refine our master key filter set. While our 8 filters demonstrate impressive performance, systematic optimization might further improve their effectiveness or reduce their number. The observed constraint on filter diversity invites us to rethink the fundamental principles governing these architectures, potentially leading to insights about the complementary roles of depthwise spatial filters and pointwise channel mixing layers and opening opportunities for novel architecture designs.

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

# A Technical Appendices and Supplementary Material

## A.1 Greedy Algorithm pseudo code

---

**Algorithm 1** Greedy Filter Selection

---

**Require:** $F \in \mathbb{R}^{C \times k^2}$: original network filters
**Require:** $F' \in \mathbb{R}^{m \times k^2}$: candidate filter pool, $m \approx 50$
**Ensure:** $S \subseteq F'$: minimal filter set
 1: Initialize $S \leftarrow F'$
 2: **repeat**
 3:     **for** each $f'_i \in S$ **do**
 4:         Approximate $F$ using $S \setminus \{f'_i\}$ with linear shifts: $\tilde{F}_i = \arg\min \| F - (a f'_{k \neq i} + b) \|$
 5:         Compute accuracy $\mathrm{Acc}(\tilde{F}_i)$
 6:     **end for**
 7:     Take the least important filter: $f^* \leftarrow \arg\min_{f'_i \in S} |\mathrm{Acc}(S) - \mathrm{Acc}(\tilde{F}_i)|$
 8:     Remove the least important filter: $S \leftarrow S \setminus \{f^*\}$
 9: **until** Elbow point detected in accuracy curve
10: **return** $S$

---

## A.2 Autoencoder Model Bank Details

To construct a comprehensive and diverse model bank for filter distillation, we collected pretrained convolutional neural networks spanning multiple architecture families and training configurations. Our goal was to capture a wide range of depthwise convolutional filter characteristics representative of both lightweight and large-scale models.

The bank includes models from the ConvNeXtV2 [34], ConvNeXt [24], MogaNet [18], HorNet [27], and ConvMixer [32] families, each trained on either ImageNet-1k or ImageNet-22k (and their 22k→1k fine-tuned variants) at multiple input resolutions. All models were obtained from publicly available checkpoints released by their respective authors. From each model, we extracted all depthwise convolutional kernels of size $7 \times 7$.

**Model Families and Variants:**

The **ConvNeXtV2** family includes the following variants: *atto-224-1k*, *femto-224-1k*, *pico-224-1k*, *nano-224-1k*, *nano-224-22k*, *nano-384-22k*, *tiny-224-1k*, *tiny-224-22k*, *tiny-384-22k*, *base-224-1k*, *base-224-22k*, *base-384-22k*, *large-224-1k*, *large-224-22k*, *large-384-22k*, *huge-224-1k*, *huge-384-22k*, and *huge-512-22k*.

The **ConvNeXt** family includes: *tiny-224-1k*, *tiny-224-22k_1k*, *tiny-384-22k_1k*, *tiny-224-22k*, *small-224-1k*, *small-224-22k_1k*, *small-384-22k_1k*, *small-224-22k*, *base-224-1k*, *base-384-1k*, *base-224-22k_1k*, *base-384-22k_1k*, *base-224-22k*, *large-224-1k*, *large-384-1k*, *large-224-22k_1k*, *large-384-22k_1k*, *large-224-22k*, *xlarge-224-1k*, *xlarge-224-22k_1k*, *xlarge-384-22k_1k*, and *xlarge-224-22k*.

The **MogaNet** family includes: *tiny-224-1k*, *tiny-224-1k*, *tiny-256-1k*, *small-224-1k*, *base-224-1k*, *large-224-1k*, and *xlarge-224-1k*.

The **HorNet** family includes: *tiny-224-1k*, *small-224-1k*, *base-224-1k*, and *large-224-1k*.

Finally, the **ConvMixer** family includes: *512-20_layer-1k*, *768-32_layer-1k*, and *768-initialized\**.

In total, the model bank comprises more than one million filters from 49 pretrained models across five architecture families.

## A.3 Filter Type Proportions

To illustrate the proportions of filters assigned to each of our eight filter types, Figure 6 shows the corresponding percentages for ConvNeXtV2 B and L, as well as HorNet T and B. An interesting observation at first glance is that each model family exhibits fairly consistent proportions of each filter type in its trained filters, regardless of model size. Depending on the architecture, the most

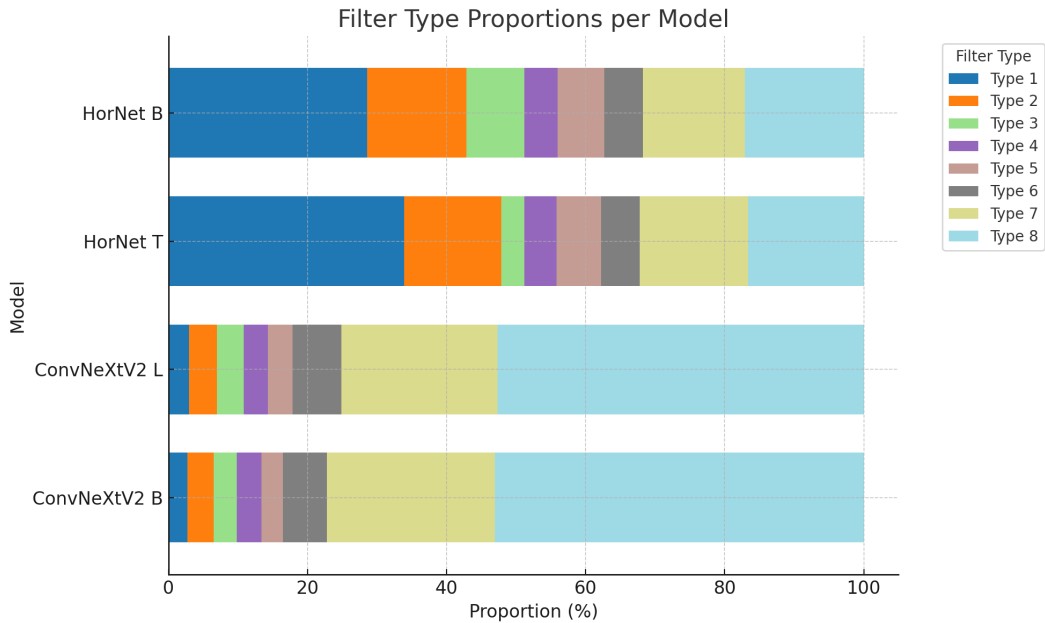

Figure 6: Proportions of the eight filter types across ConvNeXtV2 and HorNet models. Each model family shows consistent internal distributions.

prevalent filter type differs. For instance, in ConvNeXtV2, the most frequent was the Gaussian filter (Type 8), whereas in HorNet, it was Type 1. As shown, ConvNeXtV2 has similar proportions within its family but distinct distributions compared to the HorNet family. Despite the lower percentages of filter types 1–4 in ConvNeXtV2, they remain essential for model accuracy. Experiments in which these four types were removed resulted in a noticeable drop in accuracy.

### A.4 Filter Type Approximation Quality

Figure 7 shows the box plots of the distances between filters and their assigned filter types. For reference, a random filter and its corresponding distance are also included. As the plots indicate, the mean distance for all filter types is lower than that of the reference, confirming meaningful type-specific clustering. Notably, Filter 7 exhibits the lowest mean distance while also accounting for more than 20% of all filters in the ConvNeXtV2 models (See Figure 6.)

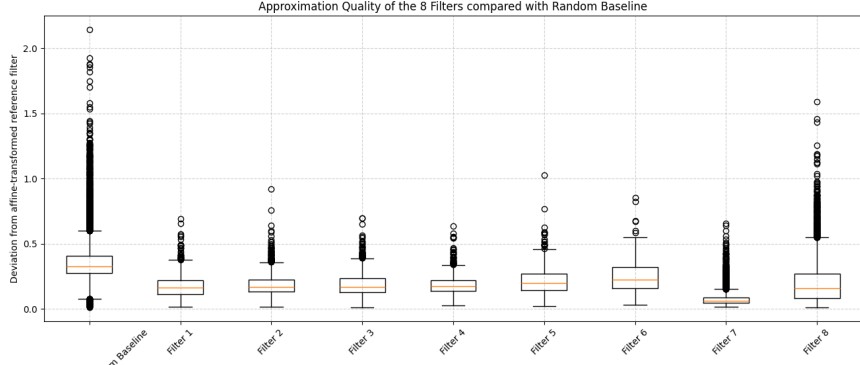

Figure 7: Box plots of distances between filters and their assigned filter types. All filter types show lower mean distances than the random reference, with Filter 7 exhibiting the lowest mean and strongest clustering.

## A.5 Experimental Settings

Table 4: Training (t) and fine-tuning (ft) hyperparameters used in Section 4.2 experiments for ConvNeXtv2 Tiny model, taken from [34].

| config | value |
|---|---|
| optimizer | AdamW |
| base learning rate | 8e-4 |
| weight decay | 0.05 |
| optimizer momentum | $\beta_1, \beta_2 = 0.9, 0.999$ |
| layer-wise lr decay [7, 6] | 0.9 |
| batch size | 1024 |
| learning rate schedule | cosine decay |
| warmup epochs | (t) 40, (ft) 3 |
| training epochs | (t) 300, (ft) 100 |
| augmentation | RandAug (9, 0.5) [9] |
| label smoothing [29] | 0.1 |
| mixup [38] | 0.8 |
| cutmix [37] | 1.0 |
| drop path [17] | 0.2 |
| head init [31] | 0.001 |
| ema | 0.9999 |

Table 5: Training (t) and fine-tuning (ft) hyperparameters used in Section 4.2 experiments for ConvNeXtv2 Pico model, taken from [34].

| config | value |
|---|---|
| optimizer | AdamW |
| base learning rate | 2e-4 |
| weight decay | 0.3 |
| optimizer momentum | $\beta_1, \beta_2 = 0.9, 0.999$ |
| layer-wise lr decay [7, 6] | 0.9 |
| batch size | 1024 |
| learning rate schedule | cosine decay |
| warmup epochs | 0 |
| training epochs | (t) 600, (ft) 100 |
| augmentation | RandAug (9, 0.5) [9] |
| label smoothing [29] | 0.2 |
| mixup [38] | 0.3 |
| cutmix [37] | 0.3 |
| drop path [17] | 0.0 |
| head init [31] | 0.001 |
| ema | 0.9999 |

## A.6 Training Curves

Figure 8 shows the training and validation losses for CONVNEXTV2-TINY trained with the original script and with the eight convolutional filters frozen ("8-filters"). The constrained model converges slightly more slowly and maintains a higher training loss throughout (e.g., final train loss $\approx 2.84$ vs. 2.78 for the original). However, the validation losses converge to nearly the same level (final val loss $\approx 0.750$ vs. 0.740), indicating comparable generalization despite reduced train-time flexibility.

## A.7 8 Master Key Filters

In Figure 4 we provide the full numerical values of the 8 discovered universal filters, each of size $7 \times 7$. These filters were derived from a greedy search over encoded depthwise filters, as described in Section 3.1, and used in all experimental evaluations (Sections 4.1 and 4.2).

Table 6: Information of Datasets used in the study and sample sizes, in training set size descending order.

| Dataset | Classes | Train Samples | Test Samples |
|---|---|---|---|
| ImageNet | 1000 | 1.2 million | 50,000 |
| CIFAR-10 [16] | 10 | 50,000 | 10,000 |
| STL-10 [8] | 10 | 5,000 | 8,000 |
| Oxford-IIIT Pets [26] | 37 | 3,680 | 3,369 |
| Oxford 102 Flowers [25] | 102 | 2,040 | 6,149 |

Table 7: Training hyperparameters used in Section 4.2 experiments. The setting is taken from ConvNeXt [24].

| config | value |
|---|---|
| optimizer | AdamW |
| base learning rate | 4e-3 |
| weight decay | 0.05 |
| optimizer momentum | $\beta_1, \beta_2 = 0.9, 0.999$ |
| batch size | 4096 |
| training epochs | 300 |
| learning rate schedule | cosine decay |
| warmup epochs | 50 |
| warmup schedule | linear |
| layer-wise lr decay | None |
| randaugment | (9, 0.5) |
| mixup | 0.8 |
| cutmix | 1.0 |
| random erasing | 0.25 |
| label smoothing | 0.1 |
| layer scale | 1e-6 |
| head init scale | None |
| gradient clip | None |

## A.8 Experimental Compute Resources

We used 2 NVIDIA TITAN RTX GPUs for experiments on other datasets. For ImageNet training and fine-tuning we used 8 NVIDIA TITAN RTX GPUs.

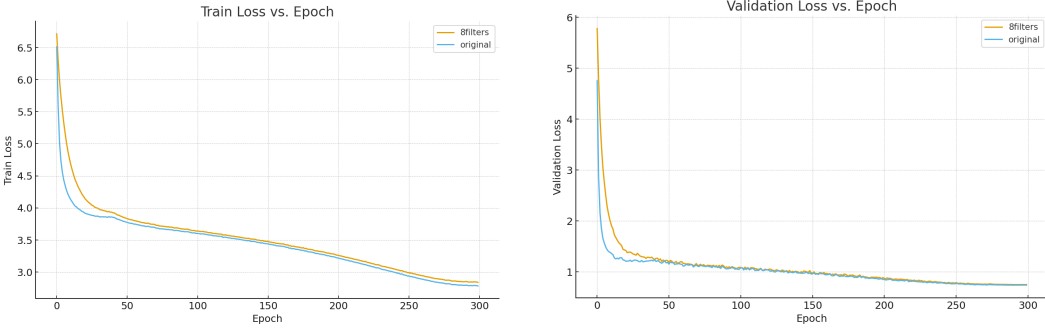

(a) Train loss vs. epoch (8-filters vs. original).

(b) Validation loss vs. epoch (8-filters vs. original).

Figure 8: Training curves for ConvNeXtV2-Tiny with the original training setup and with eight frozen filters ("8-filters").

### (a) Filter 1

| | | | | | | |
|---|---|---|---|---|---|---|
| -0.01 | -0.02 | -0.01 | -0.00 | -0.01 | -0.02 | -0.01 |
| -0.02 | -0.02 | -0.00 | 0.00 | -0.01 | -0.02 | -0.01 |
| -0.01 | -0.02 | 0.01 | -0.11 | -0.00 | -0.01 | -0.01 |
| -0.03 | -0.05 | -0.09 | -0.23 | -0.06 | -0.05 | -0.03 |
| -0.03 | -0.06 | 0.02 | 0.94 | 0.04 | -0.06 | -0.03 |
| -0.02 | -0.02 | 0.00 | 0.12 | 0.01 | -0.02 | -0.02 |
| -0.02 | -0.02 | 0.01 | 0.09 | 0.00 | -0.02 | -0.02 |

### (b) Filter 2

| | | | | | | |
|---|---|---|---|---|---|---|
| -0.00 | -0.01 | -0.02 | -0.05 | -0.04 | -0.02 | -0.00 |
| -0.02 | -0.02 | -0.02 | -0.04 | -0.03 | -0.02 | -0.03 |
| -0.02 | -0.00 | -0.01 | -0.06 | 0.06 | -0.01 | -0.01 |
| 0.00 | 0.04 | -0.06 | -0.46 | 0.85 | 0.13 | 0.07 |
| 0.00 | 0.01 | 0.01 | -0.12 | 0.07 | 0.02 | 0.01 |
| -0.01 | -0.01 | -0.01 | -0.05 | -0.03 | -0.01 | -0.01 |
| 0.00 | -0.01 | -0.01 | -0.04 | 0.00 | -0.01 | 0.00 |

### (c) Filter 3

| | | | | | | |
|---|---|---|---|---|---|---|
| -0.03 | -0.02 | -0.02 | 0.07 | -0.02 | -0.03 | -0.03 |
| -0.03 | -0.02 | 0.01 | 0.14 | 0.01 | -0.02 | -0.03 |
| -0.03 | -0.04 | 0.10 | 0.88 | 0.11 | -0.05 | -0.04 |
| -0.02 | -0.02 | -0.08 | -0.36 | -0.09 | -0.03 | -0.03 |
| -0.02 | -0.00 | -0.05 | -0.14 | -0.05 | -0.01 | -0.02 |
| -0.01 | -0.01 | 0.01 | 0.01 | 0.00 | -0.01 | -0.01 |
| -0.01 | 0.00 | 0.00 | 0.01 | 0.01 | 0.00 | -0.00 |

### (d) Filter 4

| | | | | | | |
|---|---|---|---|---|---|---|
| -0.04 | -0.03 | -0.02 | -0.01 | 0.00 | -0.00 | -0.01 |
| -0.04 | -0.01 | -0.04 | -0.01 | 0.03 | 0.01 | -0.01 |
| -0.01 | 0.00 | 0.03 | -0.05 | 0.00 | 0.02 | 0.01 |
| 0.04 | 0.08 | 0.87 | -0.35 | -0.30 | -0.00 | -0.00 |
| -0.02 | 0.00 | 0.05 | -0.01 | -0.05 | -0.00 | -0.00 |
| -0.03 | -0.01 | -0.01 | 0.00 | 0.00 | 0.00 | -0.02 |
| -0.04 | -0.02 | -0.01 | -0.01 | -0.00 | -0.00 | -0.00 |

### (e) Filter 5

| | | | | | | |
|---|---|---|---|---|---|---|
| 0.05 | 0.02 | 0.04 | 0.01 | -0.04 | -0.02 | -0.07 |
| 0.04 | 0.03 | 0.05 | 0.02 | -0.02 | -0.01 | -0.07 |
| 0.10 | 0.09 | 0.19 | 0.02 | -0.17 | -0.06 | -0.09 |
| 0.20 | 0.20 | 0.54 | -0.03 | -0.53 | -0.20 | -0.22 |
| 0.09 | 0.08 | 0.19 | 0.01 | -0.22 | -0.09 | -0.11 |
| 0.04 | 0.03 | 0.07 | 0.01 | -0.04 | -0.02 | -0.07 |
| 0.05 | 0.02 | 0.05 | -0.00 | -0.04 | -0.03 | -0.07 |

### (f) Filter 6

| | | | | | | |
|---|---|---|---|---|---|---|
| -0.07 | -0.05 | -0.08 | -0.16 | -0.07 | -0.04 | -0.06 |
| -0.03 | -0.01 | -0.06 | -0.14 | -0.04 | 0.00 | -0.03 |
| -0.03 | -0.04 | -0.22 | -0.47 | -0.22 | -0.03 | -0.04 |
| -0.01 | -0.01 | 0.01 | 0.02 | 0.01 | -0.00 | 0.00 |
| 0.02 | 0.03 | 0.20 | 0.68 | 0.20 | 0.02 | 0.03 |
| -0.00 | 0.02 | 0.06 | 0.16 | 0.05 | 0.01 | 0.01 |
| 0.02 | 0.03 | 0.05 | 0.14 | 0.06 | 0.03 | 0.04 |

### (g) Filter 7

| | | | | | | |
|---|---|---|---|---|---|---|
| -0.01 | -0.01 | -0.01 | -0.02 | -0.02 | -0.00 | -0.01 |
| -0.01 | -0.00 | -0.02 | -0.05 | -0.01 | -0.00 | -0.01 |
| -0.01 | -0.01 | -0.04 | -0.06 | -0.05 | -0.01 | -0.01 |
| -0.01 | -0.03 | -0.01 | 0.98 | -0.02 | -0.04 | -0.02 |
| -0.01 | -0.01 | -0.05 | -0.07 | -0.06 | -0.02 | -0.02 |
| -0.01 | -0.01 | -0.01 | -0.05 | -0.01 | -0.00 | -0.01 |
| -0.01 | -0.01 | -0.02 | -0.03 | -0.02 | -0.01 | -0.01 |

### (h) Filter 8

| | | | | | | |
|---|---|---|---|---|---|---|
| -0.04 | -0.04 | -0.04 | -0.02 | -0.04 | -0.03 | -0.04 |
| -0.04 | -0.03 | -0.04 | -0.04 | -0.04 | -0.03 | -0.04 |
| -0.04 | -0.04 | -0.02 | 0.16 | -0.01 | -0.04 | -0.04 |
| -0.02 | -0.04 | 0.16 | 0.92 | 0.15 | -0.04 | -0.02 |
| -0.04 | -0.05 | -0.03 | 0.15 | -0.03 | -0.05 | -0.04 |
| -0.04 | -0.03 | -0.04 | -0.04 | -0.04 | -0.03 | -0.04 |
| -0.04 | -0.04 | -0.04 | -0.02 | -0.04 | -0.03 | -0.04 |

Figure 9: The 8 filters.

