# OpenReview forum: "The Quest for Universal Master Key Filters in DS-CNNs"
_NeurIPS.cc/2025/Conference — NeurIPS 2025 poster_

### Official Review · Reviewer_kyfb · 2025-06-07

**Clarity:** 4
**Significance:** 3
**Originality:** 3
**Rating:** 5
**Confidence:** 4

**Summary:**

This work focuses on the Master Key Filters Hypothesis for Depthwise Separable Convolutional Neural Networks (DS-CNNs). It improves on previous works by showing that the trained filters can be replaced by affine transformations of a small set of base filters, with little harm to the model accuracies. These filters are found by training an autoencoder on a collection of existing trained models' filters and then drawing samples with it. This work also draws theoretical insights by comparing these base filters with Gaussian filters and shows they coincide. Extensive numerical experiments are done on ImageNet, CIFAR-10, and a few others using different sizes of ConvNeXt models. Through these experiments, this work freezes the filters to be the few base filters it has chosen and arrives at a comparable accuracy after training for certain epochs.

**Questions:**

1. What is $n$ in Eq (1)?
2. The authors find the affine transform of a single base filter for each trained filter. What is different if one finds a linear combination of all base filters for each trained filter? Since the authors use only a small set of base filters, I think the number of coefficients is still acceptable.
3. From a perspective different from saving memory, can one facilitate training by first freezing the 8 base filters for a few epochs and then allowing them to be trained for a few epochs? It seems from Table 3 that frozen filters give faster convergence in 300 epochs, compared to fully trainable filters.
4. Is there a connection to overparametrizations? See Strengths.
5. Is there also a connection to the recent research on the implicit low-rank bias of deep CNNs [1,2]? This line of research essentially shows, when trained with regularization, deep CNNs tend to have most of their filters aligned with the pooling (up to linear transformation) so the information kept by the trainable filters is also kept by pooling. In this case, most filters are linear transformations of each other. In particular, it raises the question if the base filters found in this work happen to allow most information to pass under any (or the popular) poolings.

*I would appreciate any answers the authors have to these questions.

References:

[1] Súkeník, Peter, Marco Mondelli, and Christoph Lampert. "Neural Collapse versus Low-rank Bias: Is Deep Neural Collapse Really Optimal?." arXiv preprint arXiv:2405.14468 (2024).

[2] Wen, Yuxiao, and Arthur Jacot. "Which Frequencies do CNNs Need? Emergent Bottleneck Structure in Feature Learning." International Conference on Machine Learning. PMLR, 2024.

**Ethical Concerns:**

["NO or VERY MINOR ethics concerns only"]

**Final Justification:**

The authors have addressed my questions to a satisfactory extent. I believe there is valuable insight in the proposed "basis" filters from a theoretical perspective, despite that its practical benefits are not yet clear. I increase my score to Accept conditioned on that the authors include/fix what has been discussed during the rebuttal.

However, I will not be able to argue against other reviewers' more negative reviews, because there are grounds for rejection: this work indeed does not have a well-rounded set of experiments to support its more theoretical findings. The boxplot I mentioned during rebuttal is one example, but may not be exhaustive.

**Limitations:**

Yes.

**Quality:**

4

**Strengths And Weaknesses:**

Strengths:
- The results presented are inspiring and practically beneficial.
- The experiments are done extensively on a number of datasets.
- The paper is well written.
- The accuracy loss incurred by filter pruning (in Table 1) is less when the model is larger, which is good since large models are the motivation for pruning. In particular, this trend seems to have a connection with overparametrization and the NTK results e.g. [1,2], in which case large MLPs find good parameters close to their initializations. The trend in Table 1 may imply the larger the model is, the less it needs to diverge from the base filters. This may be an interesting connection to make, if I am not mistaken here.

Weaknesses:
- No statistical confidence is given. The authors argue that running experiments on ImageNets is costly. But is it possible to run and give a confidence on the smaller datasets used in Section 4.2?
- Apart from ImageNet, the other datasets are relatively easy tasks. It remains a question whether such a nice concentration on the filters still holds under more complicated visual tasks.

Typos:
- Line 173, "a small set".
- Line 226, "Figure 4".
- Line 236, "This filter likely contributes".
- Line 244-245, missing $\sigma$ as subscript in defining gaussians (for $G_1$, $G_2$ to make sense in DoG).
- Table 3, "normal training"

References:

[1] Jacot, Arthur, Franck Gabriel, and Clément Hongler. "Neural tangent kernel: Convergence and generalization in neural networks." Advances in neural information processing systems 31 (2018).

[2] Allen-Zhu, Zeyuan, Yuanzhi Li, and Zhao Song. "A convergence theory for deep learning via over-parameterization." International conference on machine learning. PMLR, 2019.

---

> ### Author Rebuttal · Authors · 2025-07-31
>
> We want to thank the reviewer for their review and for accepting our paper. We have to mention that although our paper can be viewed as a kind of pruning, ours is more of an analytical paper and we are more focused on understanding the DS-CNN filters in this paper regardless of what the application can be.
>
> ---
> 1. Weaknesses:
> - **“confidence on the smaller datasets”**: For sure, For Flowers and Pets datasets we repeated the experiments 4 times to serve as a confidence, but we can complete them even more with confidence intervals. We repeated the CFAR10 for you and our results seem to be consistent. Each experiment was repeated 4 times, and we now report the results with error bars (mean ± standard deviation). The experiments are on-going that we will complete and update that table. Please find the confidence table below:
> | Dataset   | Setting                      | Accuracy            |
> |----------|------------------------------|--------------------:|
> | CIFAR-10 | Original                      | 96.95 ±0.10         |
> | CIFAR-10 | ImageNet Filters              | 97.07 ±0.18         |
> | CIFAR-10 | Acc with 8 unique Filters     | 96.36 ±0.13         |
>
> - **“filters still hold under more complicated visual tasks”:** This is a valid argument! We definitely should try our filter on other tasks. We would try the COCO object detection task but it takes time. Is there any task or dataset that you are interested in?
>
> ---
> 2. Typos:
> - Thank you very much. We would correct them.
> ---
> 3. Questions:
> - In both Eq (1) and Eq (2) It’s the length of the vectors, here 7×7 = 49. But I totally get where your confusion comes from! In the next lines “n” is different and it’s the number of different vectors (here number of filters we have). It’s a mistake on our part and we have to fix the notation. A genuine thank you for noticing this.
> - **“What is different if one finds a linear combination of all base filters for each trained filter?”:** First, we are not sure if we should use “affine transform” here because most of the time affine transform means matrix multiplication transform (Are we wrong?) but here we are just using a linear shift (ax+b) where “a” and “b” are just scalars. Second, about linear combinations, we should remember how powerful linear combinations are. To give you a simple example, linear combination of just 9 random 3×3 filters would span the space of all possible 3×3 filters because 3×3 filters have 9 dimensions. But that is not the case for linear shifts. So mathematically, if we show the filters are converging to linear shifts, we are finding a far greater constraint to the space of filters compared to if we’ve used linear combination. \
> **TLDR:** Using linear combinations would definitely increase the accuracy, but the scientific importance of it is very low compared to linear shifts.
> - You are completely right. Actually we are adding a loss comparison chart that would show that. For fast convergence, we recommend initializing your DS-CNN models with these filters; they do make a real difference for the starting epochs.
> - Sorry, we thought about it a lot, our guess would say “Yes”; however, we couldn’t come up with a scientific experiment or explanation to answer this question the professional way.
> - Oh! Those papers look super interesting! Especially “Which Frequencies Do CNNs Need?” Indeed there could be a connection because each DS-CNN is mathematically a classical CNN. We have to dive into that to figure out the connection. About the “base filters” you mentioned, I wish I could send pictures in this rebuttal, but we suspect they are bases for natural images, because we could construct them by finding the eigenvectors of the covariance matrix of any natural image we tested. It would be hard to explain without figures unfortunately. But anyways, thanks for the papers.

---

> > ### Comment · Reviewer_kyfb · 2025-08-02
> >
> > I appreciate the authors' response. I think the insight in this work is already interesting, but this work would be more complete with a more information-theoretic discussion on "why these 8 filters". For example, is there some interpretable change to the output figures when we remove selected types of the 8 filters from the model? Is there max/average pooling in the current CNNs and does pooling change the 8 final filters? I also wonder why the authors do not think classical CNNs can replicate similar results, as metioned in the response to Reviewer tTfe?

---

> > > ### Author Response · Authors · 2025-08-03
> > > **Comment for Reviewer kyfb**
> > >
> > > We want to thank the reviewer for showing interest in our work and another thanks for reading our comment to other reviewers. We appreciate it a lot.
> > >
> > > 1. **"why these 8 filters"**: At this time, we only have some scientific guesses. There are some close similarities between the 8 filters we achieved with unsupervised methods and classical image processing; more importantly, the "Scale-Space Theory" we discussed in line 90. Following classical computer vision methods, we could extract some of these filters from natural images, strengthening the assumptions that these filters can be a basis for all natural images (and we could even approximated natural images using only those filters); but the problem is we only could extract filters 5 to 8, we couldn't extract filters 1 to 4, they remain an open question for us. We believe that it takes a community effort to come up with rigorous theoretical explanation for these models, and publishing this work contributes to that.
> > >
> > > 2. **is there some interpretable change**: Changes? yes, interpretable? at least we couldn't interpret. One thing we can say from our experiments is that removing any of these filters would tangibly decrease the accuracy of the model.
> > >
> > > 3. **"Is there max/average pooling in the current CNNs and does pooling change the 8 final filters?"**: No, newer models usually don't use max/average pooling between stages anymore (only they usually use one global average pooling before the classification layer). The current SOTA method is using "downsampeling" layer between stages, which itself is a convolution layer with stride=2 and kernel=2.
> > >
> > > 4. **"why the authors do not think classical CNNs can replicate similar results"**: To avoid any ambiguity that could possible arise from "similar results", we'd better answer your question with 3 different assumption:
> > > - *Do you mean if there exists a master key filter set for classical CNNs too?* There are some evidences that suggest classical CNNs are converging towards master key sets too, but we should mention that a master key filters set can theoretically contain thousands of filters, as big as all filters in the model.
> > > - *Do you mean if there exists a master key filter set as small as 8 for classical CNNs too?* We don't want to say we're sure that it's impossible, but we say we are very skeptical if such a set exists for classical CNNs. The reason is that after visualizing the filters of both CNN models and DS-CNN models, one can clearly see repetition patterns in DS-CNN filters but the same phenomenon is not visible in classical CNNs (although they have some repeating filters too, but it's not even close).
> > > - *Do you mean if our method works for classical CNNs too?*: We tried, it doesn't.

---

> > > > ### Comment · Reviewer_kyfb · 2025-08-05
> > > >
> > > > I appreciate the authors' response. I think it would also be a helpful evidence if the authors can include a more quantitative figure on how close each trained filter is to an affine transformation of one of the selected 8 filters; for example, compute a deviation from transformations of any of the 8 filters for each trained filter, and plot a boxplot to show "most" of them are well approximated. Besides, I think the wording "basis" can be misleading, as the authors emphasize on replacing each filter with *one* of the 8, not a linear combination of them--maybe "supporting filters" or something else is more appropriate.
> > > >
> > > > Overall, I have no further questions. I will consider changing my rating accordingly during the discussion period later.

---

> > > > > ### Author Response · Authors · 2025-08-05
> > > > >
> > > > > We appreciate the reviewers' engagement with us.
> > > > >
> > > > > - **“figure on how close each trained filter is, boxplot”**: This is a very interesting idea! We did it right away and we included a random filter as the reference. We have to thank you because we found something new after doing this! Our 7th filter, which constructs about 20-25% of the filters of ConvNextv2 (You can see the percentage table in our response to reviewer MAm3), is approximating beyond our expectation! This is something worthy of diving deeper in. \
> > > > > We are not allowed to add figures but we present a table of results below, and will definitely add the plot to the revised paper.
> > > > >
> > > > >
> > > > > |       |    Reference |Filter 1 | Filter 2 | Filter 3 | Filter 4 |Filter 5 | Filter 6 | Filter 7 | Filter 8 |
> > > > > |-------|----------:|---------:|---------:|---------:|---------:|---------:|---------:|---------:|---------:|
> > > > > | Mean  |   0.410888 |  0.177565 |  0.188048 |  0.179334 |  0.160989 |  0.176890 |  0.179738 |  0.088699 |  0.211128 |
> > > > > | Std   |   0.135496 |  0.092489 |  0.076252 |  0.081119 |  0.054286 |  0.077240 |  0.054603 |  0.061809 |  0.153922 |
> > > > >
> > > > >
> > > > >
> > > > > - **“I think the wording "basis" can be misleading, "supporting filters" or something else is more appropriate.”**: You are absolutely right that the term “basis” can make confusions in our paper, we are not using linear combinations at all. And your suggestion “supporting filters” sounds good! We will probably use that instead, thank you for that.\
> > > > > But just to avoid a misconception, when we said “these filters can be a basis for all natural images” *we were not talking about any methods in our paper*. We were talking about something deeper, we meant that these filters maybe be basis in a linear combination manner to construct all natural images.\
> > > > > We have to add this to our appendix. Since we think explaining it here just makes it more complex, we thought you can see it yourself just by running a short code. The code is not a deep learning code. Please take ANY black-and-white image from the internet (we recommend the standard “The original image of Cameraman”) and replace it in the code with *"your_image.jpeg"* and run it. What we’re doing is reshaping the eigenvectors of the covariance matrix of the image patches to a square filter, and you can see how close some of them can get to our discovered filters. Then we use linear combinations of those to approximate the image patch by patch, and you can see how well they can approximate any image. (we have to reference page 204 of the book [1] for this method)\
> > > > > After seeing this, please imagine repeating the same process for thousands of images and averaging the similar filters, they would be clear gaussians-related filters and derivatives. We hope we made our point clear that these filters could be fundamental to image spaces.
> > > > > ---
> > > > >
> > > > > [1] BM Romeny. Front-End Vision and Multi-Scale Image Analysis: Multi-scale Computer Vision Theory and Applications, written in Mathematica. Springer Publishing Company, Incorporated, 1st edition, 2009.
> > > > >
> > > > > ---
> > > > >
> > > > > ```python
> > > > > import numpy as np
> > > > > import matplotlib.pyplot as plt
> > > > > from skimage import io, color
> > > > > from skimage.color import rgba2rgb
> > > > > from sklearn.preprocessing import normalize
> > > > > from scipy.linalg import eigh
> > > > >
> > > > > k = 7
> > > > > A = io.imread("your_image.jpeg")
> > > > > if A.ndim > 2:
> > > > >     A = color.rgb2gray(rgba2rgb(A) if A.shape[2] == 4 else A)
> > > > > A = A.astype(float)
> > > > >
> > > > > h, w = A.shape
> > > > > patches = [A[y:y+k, x:x+k]
> > > > >            for y in range(2, h-k, k)
> > > > >            for x in range(2, w-k, k)]
> > > > > d = np.arange(k) - (k//2)
> > > > > g = np.exp(-(d[:, None]**2 + d**2) / 2)
> > > > > g /= g.sum()
> > > > >
> > > > > X = (np.array(patches) * g).reshape(-1, k*k)
> > > > >
> > > > > X -= X.mean(0)
> > > > >
> > > > > eigvecs = eigh(X.T @ X)[1]
> > > > > V = normalize(eigvecs[:, -8:].T)
> > > > > eigen_patches = V.reshape(8, k, k)
> > > > >
> > > > > fig, axs = plt.subplots(2, 4, figsize=(10, 5))
> > > > > for i, ep in enumerate(eigen_patches):
> > > > >     axs.flat[i].imshow(ep, cmap='gray')
> > > > >     axs.flat[i].axis('off')
> > > > > plt.show()
> > > > >
> > > > > c = 5
> > > > > offset = (k - c) // 2          # = 1
> > > > > cropped = eigen_patches[:, offset:offset+c, offset:offset+c]
> > > > > V = cropped.reshape(8, -1)
> > > > > patches = [A[y:y+c, x:x+c]
> > > > >            for y in range(2, h-c, c)
> > > > >            for x in range(2, w-c, c)]
> > > > > X = (np.array(patches)).reshape(-1, c*c)
> > > > > X -= X.mean(0)
> > > > >
> > > > > weights = X @ V.T
> > > > > recon_patches = (weights @ V).reshape(-1, c, c)
> > > > > recon = np.zeros_like(A)
> > > > > count = np.zeros_like(A)
> > > > > idx = 0
> > > > > for y in range(2, h-c, c):
> > > > >     for x in range(2, w-c, c):
> > > > >         recon[y:y+c, x:x+c] += recon_patches[idx]
> > > > >         count[y:y+c, x:x+c] += 1
> > > > >         idx += 1
> > > > > count[count == 0] = 1
> > > > > recon /= count
> > > > >
> > > > > plt.figure(figsize=(10, 5))
> > > > > plt.subplot(121)
> > > > > plt.imshow(A, cmap='gray')
> > > > > plt.axis('off')
> > > > > plt.subplot(122)
> > > > > plt.imshow(recon, cmap='gray')
> > > > > plt.axis('off')
> > > > > plt.show()
> > > > > ```

---

> > > > > > ### Comment · Reviewer_kyfb · 2025-08-08
> > > > > >
> > > > > > I thank the authors for providing these results. I have no further questions and will consider adjusting my score accordingly in the discussion period later.

---

### Official Review · Reviewer_tTfe · 2025-07-02

**Clarity:** 2
**Significance:** 3
**Originality:** 3
**Rating:** 3
**Confidence:** 3

**Summary:**

The paper describe a method to identify master key filters from separable CNNs, and claims that these filters mimics the receptive field of in mammalian visual systems. Furthermore, the authors show that the separable CNNs with frozen master key filters (with learnable bias) are able to achieve comparable performance as the fully trainable counterpart.

**Questions:**

**Writing**: I would strongly recommend the authors to enhance the technical precision of the writing.
  - The paper only studies depth-wise CNN, but keep writing depth-wise separable CNNs (DS-CNNs). For a 7x7 filter, a depth-wise filter has 49 parameters, but a depth-wise separate one has only 14 parameters.
  - From Line 114-117, I don't understand why centering is an issue as the filter is flattened into a vector anyway. Also, the authors write 'scaling their length to 1', but the readers would naturally think it is the filter length (7).
 - The network has 4 encoder/decoders layers, but Figure 2 only has one.
 - In Line 140, the matrix F is never defined before and never used later.
 - In Line 142, the authors write 'a linear approximation with respect to the decoded filters' --- the readers would think it is the linear combination of decoded filters.
 - Equation (1)-(3) are very standard linear regression --- there is no need to distract the audience with details.
 - The greedy algorithm is only mentioned by words, and not in maths (or pseudo code).
 - The whole Section 3.1 is three page long without subsection or delimiters.

**Method**:
  - The auto encoder-decoder model is not the most up-to-date method for unsupervised clustering.
  - I wonder if there is any specific reason why a standard VAE (or a more advanced categorical VAE) is not used in this setting.
  - Also, the proposed method naturally applies to standard CNN instead of depth-wise CNN --- does it mean the phenomenon can only be observed in depth-wise but not a standard one?

**Experiments**:
  - From my understanding, the majority of the parameters remain learnable in the experiment setup. For example, if the number of channels if 1024, the number of parameters in the depth-wise filter is only 49*1024~=50K, while the number of parameters in the fully-connected part is 1024x1024=1M --- so the parameters in the depth-wise filter is negligible.
  - I hypothesize that if you initialize the depth-wise filters randomly and remain unchanged during learning, the model is able to achieve similar results as the fully-learnable one.
  - Also, I would recommend adding experiments compressing a standard CNN in a similar manner - if the compressed CNN achieve similar result, that will make the hypothesis in the paper more convincing.

**Ethical Concerns:**

["NO or VERY MINOR ethics concerns only"]

**Final Justification:**

The rebuttal addresses part but not all of the concerns --- so I am willing to increase the score to 3.

**Limitations:**

See above.

**Quality:**

3

**Strengths And Weaknesses:**

Strengths:
1. The idea that the filters in separable CNNs mimics receptive fields in mammalian visual systems is very intriguing, which also challenges the assumptions that the filters for later layers extract are more complicated than earlier ones.
2. The flow of the paper is smooth --- from hypothesis to investigation; from results to analyzing; from observations to experiments.

Weakness:
1. The writing overall is not very clear, especially when technical precision is required (see Questions section for details). But I would like to point out an example that annoys me most here --- the paper heavily uses the term 'linear shift' without a precise description. When I first read it, I thought it means geometric shift instead of amplitude shift. Also, such mapping should be called 'affine' instead of 'linear'.
2. The methods used in the paper (both for analysis and experiments) are not the most solid ones in my opinion. For example, a variational encoder is more suitable for the scenario, but the authors choose to use a vanilla encoder-decoder.

---

> ### Author Rebuttal · Authors · 2025-07-31
>
> We thank you for your time and comments. We have carefully considered each point raised and have done our best to address them in detail below. Thank you for helping us improve the paper.
>
> ---
>
> **Weaknesses:**
>
> 1.**“'linear shift' without a precise description”**: We tried to define linear shifts as soon as line 6 of the abstract as a simple (ax + b) operation where “a” and “b” are scalars. For example (7, 7) = 2 × (3, 3) + 1. We mentioned this once again in line 264 “mathematically expressed as a(x + b).”. We would clarify this more by writing an example in the paper. This is a very simple operation and linear shift was the best term we could come up with. If you have any suggestion about using a better term we would appreciate your input. Affine transformation requires matrix multiplication to move a geometric shape, but here “a” is only a scalar.
>
> 2.VAEs are not suitable for this task. VAEs are used as a probabilistic generative model to produce new samples and generate interpolations between samples, but here we strongly do not want any new sample. And we strongly need to keep the code dimension to be 1, because then we can select limited samples (in this paper 50). But 1D latent space is much more restrictive for VAE and doesn’t make sense. We’ve spent months and tried many different models to finally come to auto encoders, I think you are underestimating the power of AE in this task to just call them “vanilla”. By the way, we would be happy for sure if we can find a better model/method to do this task. If there is any other method that is not the most solid we are happy to hear.
>
>
> ---
>
> **Writing:**
>
> - We believe the reviewer is not familiar with the literature of DS-CNNs. We are indeed studying DS-CNNs models [1], We call the convolution part of DS-CNNs “Depthwise Conv” and when it is combined with the next fully connected layer we call the whole model Depthwise Separable Convolution model. And No! 7×7 filters indeed have 49 parameters, not 14. Working on this topic for long, we really don’t know which reference you are pointing to here (49 parameter depth-wise CNN) vs (14 parameter DS-CNN). Could you please cite a model that is DS-CNNs and has 14 parameter for its 7×7 filter?
>
> - Thanks for helping us improve the paper. We would change “length” to “norm” to not make this confusion. In our experiments center-norming helped the autoencoder to encode better.
>
> - Figure 2 is just for visualization purposes not the actual model. Our model also inputs 7×7=49 parameters but we showed a 3×3=9 input, because the other one was just too big to show.
>
> - Matrix F is actually  defined in line 140, and it is used later, in line 159. Matrix F serves as a compact representation of all the depthwise filters in a layer.
>
> - “In Line 142” You are totally right in this matter this is a valid confusion that might accrue. We have to rewrite that section carefully to clarify this. Thank you for pointing that out.
>
> - We wrote in 147 that “we use linear regression” and in 149 “ This problem has a well-known solution.” But we needed to put the formula to make the other steps clear to the reader, as we say in 151 “Calculating Equations (1) can be computationally intensive” then we follow by our own method to reduce the computation by normalising the candidate filters first. We don’t think this is trivial for all readers.
>
> - “greedy algorithm” At first we considered this as trivial, but now we would add pseudo code to the appendix to improve the clarity.
>
> - “ three pages long without subsection”: thank you for pointing this out. We would try our best to make that long section more clear.
> ---
>
> **Method:**
> - We have addressed this above.
>
> - “Does it mean the phenomenon can only be observed in depth-wise but not a standard one?” This is an excellent question that we can write a lot about. There is evidence that classical CNNs are converging towards some “Master Key filters” too, but the evidence for DS-CNN filters is much stronger. On the other hand, unlike classical CNNs, DS-CNN filters have been shown to follow gaussian patterns, and that was our motivation for this paper to see how much can we shrink this pool? Although maybe “Master Key Hypothesis” is true for classical CNNs, we don’t think we ever can find a set as small as 8 filters for them.
>
> ---
>
> **Experiments:**
> - This is completely true and valid, and we did not claim we do parameter reduction in this paper. This is an analytical paper for scientific understanding.
>
> - This hypothesis is wrong. First, we already put a random experiment in Table 1, in which we showed that the trained filters are converging to a linear shift of one of our 8 filters out of nearly infinite possible filters in the space. Second, for you, we conducted the experiment of Table 2 with 8 random filters, and after 80 epochs we stopped the experiment due to low performance. The accuracy was 23.7 at epoch 80. For your reference, our accuracy was 78 at this epoch.
>
>
> - We doubt similar results can be achieved for classical CNNs, but it is worth trying, and we will explore this.
>
> ---
> [1] [Chollet, F. (2017). Xception: Deep learning with depthwise separable convolutions. Proceedings of the IEEE Conference on Computer Vision and Pattern Recognition (CVPR), 1251–1258. https://doi.org/10.1109/CVPR.2017.195]

---

> ### Comment · Area_Chair_zJZQ · 2025-08-07
> **Please engage with the rebuttal**
>
> Dear reviewer,
>
> Thank you for reviewing this paper. The authors have provided a detailed rebuttal to your review, but I cannot see any edits to your review that take the author reply into account, and also your "Final justification" is missing. *Please engage with to the rebuttal asap*. Note I flagged you as non-responsive, but this flag will be removed once you posted a satisfactory reply
>
> Thank you, the AC

---

### Official Review · Reviewer_MAm3 · 2025-07-02

**Clarity:** 4
**Significance:** 3
**Originality:** 3
**Rating:** 4
**Confidence:** 4

**Summary:**

The paper presents an investigation on the diversity of learned layers in convolutional NNs with depthwise separable convolutions (DS-CNNs). The results of this investigation show that all the learned spatial convolutional kernels are the linear transformations of 8 basic layers. Two main experimental verifications are as follows: 1) All spatial kernels in a trained DS-CNN can be exchanged for the linear transformation of a closest kernel from this set with small accuracy drop; 2) All spatial kernels can be initialized with kernels from this set, leaving only the bias trainable, with small accuracy drop.

**Questions:**

Please see **action items** in “Weaknesses”. I will be willing to accept the paper if the first three are done. Moreover, I have some smaller questions:
1. Did you think about using the discovered fact for purposes of quantization? How effective would such quantization be?
2. Are you planning to release the code? It would be nice for everyone to be able to initialize/exchange existing layers in their models with the discovered 8 classes.

**Ethical Concerns:**

["NO or VERY MINOR ethics concerns only"]

**Final Justification:**

During the rebuttal, the authors addressed W1, W2 and W4 from my initial review, and promised to add missing graphs from W3. Thus, I raise the “clarity” score to 4 and the rating to 4. Overall, I think that the achieved result is interesting, but I am not sure if it is significant enough for rating 5.

**Limitations:**

The authors adequately addressed the limitations and potential negative societal impact of their work.

**Quality:**

3

**Strengths And Weaknesses:**

**Strengths**:
1. The paper presents and experimentally verifies a novel and fascinating idea. The experimental part is convincing.
2. The process of searching for these filters is explained in detail.

**Weaknesses**:

The paper is very insightful and captivating. My comments are mostly about the lack of detailed experimental description and suggestions for new experiments:
1. The paper glosses over two important experimental details: 1) How large is the model bank which the pretrained kernels were taken from? 2) How to choose the correct kernel from these 8 for a specific layer in the model during initialization? **Action item**: please answer these questions in the paper.
2. Also, after establishing the 8 classes of filters, a question about the importance and role of each filter naturally arises. It would be nice to see some statistics: which class is the most popular in the pretrained networks? How are they situated with respect to the layer depth? **Action item**: please compute these statistics.
3. Moreover, it would be very interesting to compare the training loss dynamic with and without initialization, which was done in [1]. **Action item**: please report the graphs with training/validation loss dynamic.
4. The authors consider only kernels of size 7 by 7. Arguably, the 3 by 3 kernels are also widely used – do they have the same properties? **Action item**: please confirm experimentally what would happen if 3 x 3 convolutions would be replaced by a closest from a similar set of 8 3 x 3 convolutions.

Also, some small problems with the manuscript:
1. In my opinion, the “Related work” section is unnecessarily long. What is the connection between the current paper and scale-space theory? Moreover, I would assume that there is no need in writing formulae for depthwise convolutions, as it is a widely known concept. **Action item**: please reduce this section by removing irrelevant discussion.
2. In lines 149-150 the authors propose a well-known solution to a linear regression problem, but there is no link or citation. **Action item**: please cite a book or another article with this formula.
3. In line 197 the authors refer to the last row of Table 1, while they probably meant the row before last.

[1] Xu, Chunyu, and Hong Wang. "Research on a convolution kernel initialization method for speeding up the convergence of CNN." Applied Sciences 12.2 (2022): 633.

---

> ### Author Rebuttal · Authors · 2025-07-31
>
> We sincerely thank you for the thorough review, especially for providing clear action points. This form of review is something we truly appreciate, and we were even inspired to adopt a similar style in our own future reviews. Below we provide our responses:
>
> ---
>
> **Weaknesses:**
>
> **1.1** We gathered around 1 million filters in the bank from 54 models of different architectures, all with 7×7 depthwise filters, and with varying model sizes, dataset sizes (ImageNet-1K or 21K), and input image resolutions. We will add a summary to the main text and a comprehensive list to the appendix of the paper to cover this information.
>
> **1.2** Our observations of the proportions of filters in each model revealed that each model family has rather consistent proportions of each filter type in its trained filters, regardless of model size (more details are presented in the next action item). The proportions vary slightly between layers, but our experiments suggest that a uniform distribution per layer works similarly. We used the proportions derived from the trained models for initializing the networks.
>
> ---
>
> **2.** Depending on the architecture, the most popular filter type differs. For instance, in ConvNeXtV2, the most frequent was the Gaussian filter (Type 8), whereas in HorNet, it was Type 1. We have prepared plots showing statistics in total and per layer for each model. Unfortunately, we cannot include these here due to rebuttal rules, but we will add them to the paper and appendix. Below is an illustrative table:
>
> | Model   | Type 1   | Type 2   | Type 3   | Type 4   | Type 5   | Type 6   | Type 7   | Type 8   |
> |---------|----------|----------|----------|----------|----------|----------|----------|----------|
> | ConvNeXtV2 B | 2.77%    | 3.71%    | 3.33%    | 3.55%    | 3.08%    | 6.34%    | 24.17%   | 53.05%   |
> | ConvNeXtV2 L | 2.99%    | 3.96%    | 3.89%    | 3.50%    | 3.55%    | 6.99%    | 22.47%   | 52.64%   |
> | HorNet T | 33.89%   | 13.97%   | 3.35%    | 4.64%    | 6.35%    | 5.54%    | 15.62%   | 16.64%   |
> | HorNet B| 28.57%   | 14.25%   | 8.34%    | 4.80%    | 6.72%    | 5.56%    | 14.66%   | 17.10%   |
>
> As shown, ConvNeXtV2 has proportions similar within its family but distinct from the HorNet family. Despite the lower percentages for filter types 1–4 in ConvNeXtV2, they are essential for model accuracy. We performed experiments removing those four types, and the accuracy dropped noticeably.
>
> ---
>
> **3.** Thank you for the suggestion. For the original models, we took trained weights and accuracies directly from the official repositories, so we did not initially have the training logs. We have now started training the original models on our own GPUs and have plotted the loss dynamics for training and validation for one model, both with and without frozen initialized filters. While we cannot provide figures and plots here, we will include them in the final paper.
>
> ---
>
> **4.** DS-CNNs with 3×3 filters (such as MobileNet) also show similar patterns in their trained filters. Our autoencoder method (Figure 2) does not converge well on 3×3 filters, even though Gaussian patterns are visible. In this paper, we performed our search on 7×7 filters since we could gather a large enough bank of filters and our autoencoder converged to a reach code layer.
>
> ---
>
> **Small:**
>
> **1.** Based on our experience, not all readers are familiar with DS-CNN architectures. Regarding scale-space theory, there is a fascinating link to our paper: the theory mathematically proves that convolution with Gaussian or derivative filters is the only operation that can scale an image. This classical computer vision theory proposed the Gaussian filters we found in an unsupervised manner, years before modern deep learning. We agree this related work section can be shortened, and we will do so to make room for the new suggested details and plots.
>
> **2.** We have added the missing citation. Thank you for pointing this out.
>
> **3.** We have fixed the noted issue. Thanks again.
>
> ---
>
> **Questions:**
>
> **1.** We do not have prior experience with quantization, but the idea is interesting and our results may indeed be useful in this direction.
>
> **2.** Yes, all code will be released in a public repository. Also, the 8 filters are already provided in full in the appendix, and can be directly used for initialization or other experiments.

---

> > ### Comment · Reviewer_MAm3 · 2025-08-03
> >
> > 1. Thank you, no further comments
> >
> > 2. Is there any intuition on why there is such a difference in filter type distribution for ConvNeXt and HorNet? Moreover, it is unclear why for HorNet there is such a difference between types 1-2 and 3-4 – the latter couple is much more rare, although 3 is just symmetrical for 1 and 4 is symmetrical for 2. Do you have any ideas about it?
> >
> > 3. I will be waiting for the revision with the plots!
> >
> > 4. Do you have any ideas why the autoencoder does not converge for 3 x 3 filters? Is there any conceptual difference to 7 x 7 filters? Have you conducted hyperparameter tuning?

---

> > > ### Author Response · Authors · 2025-08-04
> > > **Official Comment for Reviewer MAm3**
> > >
> > > Thank you for engaging and showing interest in our work.
> > >
> > > ---
> > >
> > > - **"Is there any intuition on why there is such a difference in filter type distribution for ConvNeXt and HorNet?"**: There is a very interesting observation we have for the *total proportion* of these filters in DS-CNN models. For a given architecture, the total proportion of filters would be relatively the same, even if you change the kernel size. That means if ones trains ConvNext tiny on ImageNet1k dataset, these proportions would be similar to ConvNext huge trained on ImageNet21k. But when changing the architecture, these proportions drastically change and this is true for all DS-CNNs not only ConvNext and HorNet (for example see Figure 10 of [1]). We think this suggests that architecture design plays a significant role on filter distribution, but scaling the models doesn't (And this shows the proportions are a fundamental aspect of a model if you think about it).
> > > - **"it is unclear why for HorNet there is such a difference between types 1-2 and 3-4"**: That's a good question and it's one of our interests too. To give you an honest answer, we've been thinking and experimenting about the effect of architecture design on proportions for months, and we believe it's not an easy question. Answering this question probably requires community efforts and a couple of creative papers. Just for your reference, HorNet has a complex architecture, very different from ConvNext, it uses recursive convolution gates + element-wise multiplications + 2D fast Fourier transform.
> > > ---
> > > - **I will be waiting for the revision with the plots!**: Open reviewer does not allow us to add a revision for our paper too. Do you have any suggestions on how we can share the plots with you?
> > > ---
> > > - **"Do you have any ideas why the autoencoder does not converge for 3 x 3 filters?"** This is a good question with an interesting answer. Yes we actually do, experimentally. And it is not because of conceptual difference to 7 x 7 filters. Look at this part of our answer in 3. "even if you change the kernel size". What we realized is that these models approximate the continuous gaussian functions with any kernel size they have. Bigger kernel sizes give you a better approximation and in a small kernel size like 3 x 3 you loose information approximating the gaussians. This seems to make it harder for the auto encoder to extract information for encoding and thus giving us incomplete filters at the decoder. But we could solve this issues by adding the wavelet transform of the filter to as a help to the encoder part, and suddenly the gaussian patterns emerged at the decoder side. The quality is still not as good as 7 x 7 filters but the experiment shows the auto encoder main struggle is to extract information from a low resolution kernel.
> > > - **"Have you conducted hyperparameter tuning?"** Yes, not only hyperparameter tuning but also architecture searching. We've conduct more than 500 experiments only for the autoencoder.
> > >
> > > ---
> > > [1]: Neural Echos: Depthwise Convolutional Filters Replicate Biological Receptive Fields. WACV 2024.

---

> ### Comment · Reviewer_MAm3 · 2025-08-06
>
> 1. "Do you have any suggestions on how we can share the plots with you?" It seems that it is impossible this year, sorry for misleading you.
> 2. “Do you have any ideas why the autoencoder does not converge for 3 x 3 filters?” Please add your explanation to the paper.
>
> I have also read your conversation with reviewer MKLF. Please note that while they might be overreacting, your writing in rebuttal comment to them is not as neutral and polite as it should be (and as it was in your other rebuttal answers here).
> 1. Please note that in formal arguments you should not consider emotional judgments, such as “This was a surprising result for us and very exciting.”, being a good argument.
> 2. Your answers also should not get personal, as in “You clearly didn’t understand the "master key filter hypothesis” paper.” – this phrase does not add any relevant information to the comment.
> 3. Please try not to be seriously offended by the reviewer writing “So, what?” to your paper: 1) English might not be their native language, so they might not even see that as offensive 2) It is absolutely impractical to try to get back at them.
> 4. If a reviewer suggests you to change the wording when they did not understand something, it is actually an important input – if ¼ of reviewers did not understand what you wrote, then probably ¼ of other readers would not understand it as well.
>
> Overall, you addressed the points from my initial review, so I will raise my score to 4.

---

> > ### Author Response · Authors · 2025-08-06
> >
> > We thank you very much for raising your score. We appreciate your interest in our paper and the time you’ve invested on reading and reviewing our work.
> >
> > We will add a detailed explanation on the appendix, with visualizations, and including our analysis of
> > why this occurs.
> >
> > We appreciate it a lot that you’ve followed our discussions with other reviewers. Thank you for the feedback on keeping a professional tone. We didn’t intend anything personal and we apologise for any sensitive choice of words, and the tense discussion. You're right about cultural differences, we aren't native english-speaker as well, but we should have framed our sentences more professionally (we are different authors by the way). We appreciate both you and dear reviewer MKLF for your patience in maintaining the discussion with us, and helping us improve. We will make sure to use the feedback we have received to revise and improve the clarity of our manuscript's text, to make sure it is understandable by all readers.

---

### Official Review · Reviewer_MKLF · 2025-07-07

**Clarity:** 4
**Significance:** 2
**Originality:** 2
**Rating:** 2
**Confidence:** 5

**Summary:**

The paper investigates the shape of depth-wise convolution filters. Specifically, based on "Master Key Filters Hypothesis" from previous paper, the paper finds a few universal filters that can represent and replace depth-wise convolution filters in various networks. An auto-encoder in the filter domain is trained on a pre-trained filter set and utilized to find the initial 50 filters. The depth-wise convolution filters in ConvNeXt v2 models are quantized with 50 filters with two parameterizations, and reduced to 8 filters at last. The paper claims that it extends "Master Key Filters Hypothesis" to only 8 filters with meaningful performance.

**Questions:**

- "Master Key" topic is already addressed in previous papers. Considering these two papers, what is the unique novelty and contribution of the paper?
    - The Master Key Filters Hypothesis: Deep Filters Are General in DS-CNNs, AAAI 2025
    - Unveiling the Unseen: Identifiable Clusters in Trained Depthwise Convolutional Kernels, ICLR 2024
- Finding the minimum "Master Key" looks interesting. But, what are the practical benefits of this technique? Can this reduce computation or improve generalization performance?
- I can't understand how "shift" works. Does "shift" use integers or floats? Is this done by learnable parameters like a and b?
- I think the "Master Key" still has a few degree of freedom: a, b, and shifts. Does it perform better than 3x3 depth-wise convolution? Because it also reduces the # of params, I think it is worth comparing with "Master Key".
The proposed method should not be a weighted sum of "Master Keys". However, I think some points are confusing. Would you please add a formula for selecting "Master Key" $f'_c$ to clarify this?
- What is the definition of $n$ in Eq. (1) and (2)?
- ConvNeXt and Hornet are pre-LayerNorm based residual architectures. Does "Master Keys" work on architectures without pre-LayerNorm? How about non-residual architectures like MobileNetV1?

**Ethical Concerns:**

["NO or VERY MINOR ethics concerns only"]

**Final Justification:**

The rebuttal has partially addressed my concerns.

The paper still has the following weaknesses:
- No practical benefits
  - 8 filters are interesting, but computation costs and parameter reduction effects are negligible
- Limited architectures
  - The paper only covers pre-LayerNorm-based DS-CNNs: ConvNeXt, and HorNet.
  - It can't be generalized to representative DS-CNN: EfficientNet series and MobileNet series
- Finding “Master Keys” on the test set
  - It is a significant issue for fair evaluation, which can't be resolved by small experiments in rebuttal comments.

I adjust my rating to reject

**Limitations:**

It would be better to explain the architecture limitation if it is not working on non pre-LayerNorm architecture.

**Paper Formatting Concerns:**

I don't have formatting concerns.

**Quality:**

4

**Strengths And Weaknesses:**

Strengths
- Writing is clear and easy to understand
- It is interesting to replace ConvNeXt models's filters with only eight filters

Weaknesses
- The contribution overlaps with previous papers
  - "Master Key" is interesting, but has already been discussed in previous papers. It significantly limits the novelty and contribution of the paper.
  - I acknowledge that implementing the master keys as eight filters is novel and makes a contribution. But, it is a bit insufficient for me.
    - The Master Key Filters Hypothesis: Deep Filters Are General in DS-CNNs, AAAI 2025
    - Unveiling the Unseen: Identifiable Clusters in Trained Depthwise Convolutional Kernels, ICLR 2024
- Limited practical contribution
  - I don't think "Master Key" has practical value. It doesn't reduce FLOPs or improve network throughput, and the reduction in the number of parameters is negligible. But it drops the performance.
  - The experiments like Table 3 can make a practical contribution. But Table 3's settings are outdated, and improvements are not significant.
- Additional parameters `A, B, shift` for "master keys"
  - The "Master Key" introduces additional parameters to represent the filters. It is reasonable to use a few parameters to represent all filters, but I feel it conflicts with "Master Key Filters Hypothesis"
  - Actually, I think `A, B` are necessary to make performance without re-training, even when "Master Key Filters Hypothesis" is true. But I wonder how the shift works. If the shift is a two-dimensional float vector that moves "Master Keys" with bilinear interpolation, it should be mentioned and clarified in the paper.
- Limited applicable architectures
  - "Master Key" is only validated on limited architectures that use depth-wise convolution with pre-Layer Norm blocks: ConvNeXt and HorNet. I doubt the finding can be generalized to CNN without pre-LayerNorm or depth-wise convolution.
  - The paper's findings are only valid on depth-wise convolution-based architectures. Thus, it can't cover a wide range of architectures, such as ViT variants and CNN without depth-wise conv, which reduces the contribution.
  - The paper's experiments only cover pre-LayerNorm architectures. It has to be verified in depth-wise conv without pre-LayerNorm: EfficientNet V2, and MobileNetV3.
- Finding "master keys" requires evaluation on the test set
  - The paper reduces the filters based on test accuracy, limiting insights into the filter shape required for performance. It would be better to provide an alternative way to reduce filters without an actual performance test.

---

> ### Author Rebuttal · Authors · 2025-07-31
>
> We genuinely appreciate the time you have put to read and review our work.
>
> **Weaknesses:**
>
> ---
>
> **1. Contribution overlaps / “limits the novelty”**
>
> This is definitely not correct and actually surprised us. The paper starts by stating:
> > “This paper extends this hypothesis by radically constraining its scope to a single set of just 8 universal filters.”
>
> Confining a hypothesis that could accept sets of hundreds of thousands of filters to an experimentally verified *8 filters* is undeniably novel. By the same logic, any paper building on the “lottery ticket hypothesis” or any other hypothesis would not be novel.
>
> We appreciate that the reviewer acknowledged:
> > “I acknowledge that implementing the master keys as eight filters is novel and makes a contribution.”
>
> Thank you for stating this.
>
> However, regarding:
> > “But it is a bit insufficient for me.”
>
> If the implication is that after reading the previous two papers it was trivial to conclude that *all* filters could be replaced by just 8 without significant accuracy drop (Table 2), we must respectfully disagree. This was a surprising result for us and very exciting. Moreover, filters 1–4 were discovered *in this paper* and are not mentioned in prior work.
>
> ---
>
> **2. Limited practical contribution**
>
> Great practical values are a major potential of theoretical and analytical understanding, we shouldn't shut down analytical papers because they aren’t engineering papers. Despite this paper not being a practical paper, we showed our 8 filters have practical initialization values in Table 3 that they even beat ImageNet transfer learning on smaller datasets (isn’t that a surprise?). We hope the future “master Key” filters can show a much better practical value.
>
> ---
>
> **3. Additional parameters A, B, shift**
>
> We believe there is a misunderstanding about both our paper and the “Master Key Filters Hypothesis.”
>
> - First, we are not introducing additional parameters to represent our 8 filters. If you are referring to linear shifts (ax+b) in Table 1, we are showing that the filters of all those trained networks are closely converged to the linear shift of one of these 8 filters, out of the near infinite space they could converge to. And this is because DS-CNN filters are scale-free due to their architecture. And then, in Table 2, we are not using linear shifts. The master keys set we are introducing are these 8 filters.
>
> - Second, there should be a misunderstanding about the master keys hypothesis too, using parameters does not conflict with the master keys hypothesis. This hypothesis says that there exists sets (not necessarily one set) of master key filters (can be thousands of different filters) that are universal for any input data, in contrast to previous beliefs. To put it simply, it says the previous hypothesis of “specialized(data-specific) filters in deep layers” is not true.
>
>
> Regarding:
> > “I think A, B are necessary”
>
> If by A, B you mean linear shifts, they are *not* necessary. Table 2 shows that a model with only 8 unique filters (no shifts) can achieve comparable accuracy.
>
> On:
> > “bilinear interpolation”
>
> We do *not* use interpolation. In Table 1, we simply apply linear shifts (aX + b). For example, (7, 7) is a shift of (3, 3) because (7, 7) = 2 × (3, 3) + 1. In Table 1 we are showing with an unsupervised experiment that filters of all models are closely converging to a linear shift of one of those 8 filters.
>
> ---
>
> **4. Limited applicable architectures**
>
> This is a DS-CNN analysis paper, but DS-CNN models are far from “limited.” Almost all top CNN-based models today are DS-CNNs. The last state-of-the-art classical CNN we recall is ResNeXt-101 from 2017, seven years ago.
> It is not a paper about ViTs, and this is *not* a limitation, it is focus.
>
> ---
>
> **5. Finding “Master Keys” requires evaluation on the test set**
>
> To address your concern, we’ll use a validation set from the training set to perform the search for the samples. The results are the same. Actually you can repeat the same experiment on any large test set you want, the results would be similar.
>
> ---
>
> **Questions:**
>
> 1. We hope the difference between our work and prior work is now clear. If not, we ask the reviewer to show:
>    - (a) where in those papers it is demonstrated that trained filters converge close to a linear shift of one of our 8 filters;
>    - (b) where it is shown that you can train a model with only 8 unique filters and achieve comparable accuracy;
>    - (c) where 50% of our filters (filters 1–4) appear in those papers.
>
> 2. We hope future works can find practical benefits, but right now finding the minimum set is more than just interesting to us, it’s scientifically important.
>
> 3. Linear shifts are discussed in our previous discussion as (aX+b) where “a” and “b” both float scalars. We don’t learn them, we only calculate them in Table 1 with linear regression discussed in line 147.
>
> 4. As we discussed, we don’t use linear shifts for training these 8 filters, the linear shift is only for Table 1.
>
> 5. In both cases It’s the length of the vectors, here 7×7 = 49. But I totally get where the confusion comes from! In the next lines “n” is different and it’s the number of different vectors (here number of filters we have). It’s a mistake from our part and we have to fix the notation.  A genuine thank you for noticing this.
>
> 6. **On “without pre-LayerNorm”**: Our intuitional answer would be yes. If you visualize the filter of all these models you’d see clear gaussian patterns no matter the task. There is no reason for us to believe this wouldn't work on non pre-LayerNorm when we can see the filters look the same. But can we show experimental results right now? No, is it because of the LayerNorm? No! It’s because of the filter size. The models you are introducing preliminary use 3×3 filters and our autoencoder method (Figure 2) just does not converge well on 3×3 filters and we couldn’t fix it although the gaussian filters are there. That’s why we have to go with 7×7 filters. If you know any non pre-LayerNorm 7×7 DS-CNN model we would be happy to get the test right now with the same 8 filters we extracted from ConvNext-base.

---

> > ### Comment · Reviewer_MKLF · 2025-08-04
> >
> > 1. Contribution overlaps / “limits the novelty”
> >
> > I'm trying to write meaningful comments for this review. But this rebuttal comment frustrates me.
> >
> > What’s the basis for novelty in the rebuttal? Your surprise?
> > Is it enough to say that my opinion is incorrect, and I can't understand the value of a hypothesis paper?
> >
> > This rebuttal makes me feel disrespected entirely as a reviewer.
> >
> > I agree with the value of the "Master key hypothesis". It is an interesting paper, but I don't think the number of filters makes it interesting. The important point is that DS conv filters are redundant and can be reduced.
> >
> > This paper reduces the number of keys to 8 filters. So, what? It is not a practical subject like pruning and quantization. The reduced filters do not automatically make a contribution. You have to explain and argue the exact value of your method. Without additional explanation, I consider it an incremental paper that simply reduces the number of filters in "Master key hypothesis".
> >
> >
> > 2. Limited practical contribution
> >
> > The transfer learning is not enough to claim a practical contribution, especially when there are no reference numbers. But, it is not significant because the paper is not a practical paper. It's just a weakness.
> >
> > 3. Additional parameters A, B, shift
> >
> > Reviewer tTfe also raises this concern.
> > I would recommend changing the words.
> >  `linear shift` doesn't mean aX + b in general computer vision.
> >
> > I have more questions on this. As I understand, the two scalars `a, b` are calculated for each filter. For example, when replacing DS-conv with `d=256`, you need 256 `a, b`. Is this correct? Whether this is learnable or not is not important; I'm concerned about the degree of freedom.
> >
> > In line 116 and 129, you apply `centering` to the filters. Does this mean zero-mean over 7x7 or geometric centering?
> >
> > I'm confused because the words `shift` and `centering` direct geometric transforms, but you argue they don't. Would you confirm that no geometric transforms are used for replacing and preparing the filters?
> >
> >
> > 4. Limited applicable architectures
> >
> > It is a weakness that the paper can't cover non-DS-CNN
> > Also, as I mentioned, the paper doesn't cover non-pre-norm DS-CNN: EfficientNet V2, and MobileNetV3.
> > I still doubt the general applicability of the paper.
> >
> > 5. Finding “Master Keys” requires evaluation on the test set
> >
> > Using the test set is a significant problem.
> >
> > Also, it would strongly enhance the contribution if the paper used a metric other than accuracy to select 8 filters.
> > If the filters are selected based on a non-accuracy metric, this might give huge insight into the importance of filters in network design.

---

> > > ### Author Response · Authors · 2025-08-04
> > > **Official Comment to Reviewer MKLF, Part 2**
> > >
> > > # PART 2
> > >
> > > 3. Additional parameters A, B, shift
> > > - **“I would recommend changing the words”**: We try our best. Are there any recommendations you would have?
> > > - **“two scalars a, b are calculated for each filter.”** only in Table 1, not the rest of the paper. And in table 1 each trained filter of DS-CNNs is approximated with (aX+b) where X is only one of our 8 filters. The degree of freedom is a plane in the space, which is coming from DS-CNN architecture, so it’s a redundancy we showed can be eliminated in the rest of the paper.
> > > - **“Centering”**: Yes centering means zero-mean, which is a very small subset of geometric transforms. geometric transforms contain deforming, full-space matrix multiplication, which we do not use.
> > > 4. Limited applicable architectures
> > > - **"Limited applicable architectures”** The focus area of this paper is DS-CNNs which are the SOTA non-transformer vision models. Also regarding limitedness, for your reference, HorNet has a complex architecture, very different from ConvNext, it uses recursive convolution gates + element-wise multiplications + 2D fast Fourier transform, but still our results were applicable.
> > > 5. Finding “Master Keys” requires evaluation on the test set
> > > - **“metric other than accuracy”** To fully resolve this concern of yours, we have performed a new experiment detailed below. \
> > > We separated 100 random samples from each of the classes of the ImageNet training set. We then used this new set as our evaluation set for the greedy search on the 50 filter samples. We followed the same steps as our previous search, and the search resulted the same 8 filters. We will add this experiment and the final 8 filters side-by-side in the revision of our paper.
> > > - **“This might give huge insight into the importance of filters in network design.”** Thanks for stating this potentiality for practical uses. I hope you change your mind that reducing thousands of filters to 8 explainable gaussian-like filters is incremental work.
> > >
> > > ---
> > >
> > > In the final paragraph, we want to say we know you happened to have disagreements with us on a work that we are passionate about its results, and this can make the discussion tense, but this is not our intention. We just want to make a common ground, remove misunderstandings (from both sides), and create a fair judgment about our work. Thank you very much for engaging.

---

> > > > ### Comment · Reviewer_MKLF · 2025-08-05
> > > >
> > > > Thank you for your response.
> > > > I apologize for the wording in my previous response.
> > > >
> > > > 1. Contribution overlaps / “limits the novelty”
> > > >
> > > > It looks like we have different views on the "master key filter hypothesis.” I understand that my perspective is not aligned with yours. But, could you give me information from my viewpoint rather than explaining yours?
> > > >
> > > > The community has diverse perspectives, and people will interpret a paper’s value in different ways. Even if you are the author of previous works, that doesn’t mean your view on novelty and value is the only valid one.
> > > >
> > > > I know that you don't agree with my view, and I also do not agree with your perspective on novelty points.
> > > > I think it might be hard to change one's mind to reach a consensus.
> > > > But is it necessary to align the view?
> > > >
> > > > My initial rating is in the borderline area.
> > > > Even though the view differs, I wrote that "is novel and makes a contribution."
> > > > I just want information that leverages differences from previous papers.
> > > >
> > > > So, would you please specify the difference from the previous papers? What are the most significant points? How many filters are used in previous work? or are the global filters that can be shared over different architectures more critical?

---

> > > > > ### Author Response · Authors · 2025-08-05
> > > > >
> > > > > Thank you for engaging with us and taking the time for discussion, we genuinely appreciate it. We fully understand you, it’s not an easy task to change views. We just want to try to have a fruitful discussion.
> > > > >
> > > > > ---
> > > > >
> > > > > We believe we can answer these question clearly:
> > > > > - **“How many filters are used in previous work?”** All filters in the model. Every single one of them. In fact, Master Key Filters Hypothesis is a generalization paper. We try to make this more clear in our next response.
> > > > > - **“​​Are the global filters that can be shared over different architectures more critical?”** Exactly, the point of the Master Key Filters Hypothesis paper is that the CNN models are probably converging to global or universal filter sets that are even transferable cross-architecture. In that paper, filter sets means “set of all filters inside the model”, redundancy is not discussed or even cared about, all filters inside DS-CNN models can be unique, they only need to be “universal” which means they should be dataset-independant (or in very simple words, if we train the same architecture on two completely different datasets, there is a chance that both models converge to the same filters, this is the hypothesis).\
> > > > > That paper is a direct response to the highly influential work (Yosinski et al. 2014) which argues that deep CNN filters are not general but they are dataset-specific (therefore a universal filter set can not exist). Master Key paper argues otherwise, deep filters are not dataset-specific, they are general. And tries to counter that 2014 paper by repeating the exact experiments they had, in both DS-CNNs and classical CNNs and arguing their poor results were due to low quality models of that time *(See Figure 3, and Figure 5 of MKFH paper)*.\
> > > > > That’s why if you look at the experiments of MKFH, they are all-filters transfer experiments *(e.g. Table 3, 6, and 7)*, they transfer all filters because the point is not redundancy inside models, the point is that the set of all filters within models are converging to a universal set that works with any dataset.\
> > > > > In our paper though, we are asking if there is a redundancy within model filters. Now if you think about it, our current paper is possible if MKFH holds true. Because if (Yosinski et al. 2014) would be right, then you won’t be able to find a limited set of universal filters, because then deep filters would be dependant on the dataset you are training the model on, and they change by dataset, and universal filters won't exist for them.
> > > > > - **“the difference from the previous papers?”** We can list that very concisely as this:
> > > > >    - This is the first paper to deal with filter redundancy and asking to what extent we can define their space. You can read that in the section “ Do We Need Thousands of Distinct Filters?” as the starting motivation of this paper.
> > > > >    - This paper shows that the DS-CNN filters are converging close to (aX+b) of one of the 8 filters (Table 1). This is new information.
> > > > >    - The filters 1, 2, 3, 4 that are extracted with our unsupervised methods are new, they are not discovered in the previous DS-CNN works.
> > > > >    - Our unsupervised method is new too. The idea of using greedy search and linear approximation to find a limited set of filters is something we came up with after exploring many different methods.
> > > > > And Table 2, achieving comparable results using 8 filters is new. We don’t think anything even near this has been shown or even discussed before.
> > > > > - Finally, the most important take of this paper for us, is that among all possible filters, DS-CNNs are converging close to a very small well-defined space. Given that deep neural networks are optimization machines, this suggests maybe there is a clear mathematical solution to the optimization problem of CNN filters. This is scientifically of high value for us, and we suspect these filters can be fundamental filters of all natural images, we hope future works can connect the dots mathematically. About this matter, we are going to add a part to the appendix of this paper, if you are interested you can see the short code we gave to the reviewer kyfb and run it yourself.
> > > > >
> > > > > ---
> > > > >
> > > > > At the end, we want to thank you again for spending the time to read, review, and discuss with us. We hope we can revise the text in our paper to better reflect our novelties and distinctions to prior work, and we thank you for helping us improve our paper.

---

> ### Author Response · Authors · 2025-08-04
> **Official Comment to Reviewer MKLF, Part 1**
>
> # PART 1
>
> We first want to say that we did not intend any disrespect towards the reviewer, and if in any way our responses have caused this feeling, we sincerely apologize. Before responding to your further concerns, we would also like to point out that in the peer-review process it is expected that both of us are “peers”. Disagreeing and expressing what we believe is right is not disrespect, and we actually appreciate disagreements. In this regard however, expressions like “So what?” are condescending, or using terms like “frustrated” are disrespectful, which we never expressed in our responses. \
> We want to have a meaningful, peer-to-peer discussion, freely expressing what we regard as facts, to come closer to a common ground.
>
> ---
>
> 1. Contribution overlaps / “limits the novelty”
> - **“The important point is that DS conv filters are redundant and can be reduced.”**: We believe there is a misunderstanding around the "master key filter hypothesis” paper. That paper is not about filter redundancy, it says nothing about it. Actually this hypothesis lets all filters of a DS-CNN model be different. What that paper is about, is filter generalization. Previously, it was believed that the deep filters of CNN models are not general and they are dataset-dependent, the "master key filter hypothesis” paper hypothesised otherwise, all filters are general and there exist universal sets for CNN models. But such a set can be as large as all filters of a model, no redundancy is needed. \
> In contrast, the important point that you are mentioning as a contribution of that paper, is one of the most important contributions of our paper! You can see it for yourself in our paper, look at line 103, our bold title is “Do We Need Thousands of Distinct Filters?” This is actually the single motivation of this paper! Exactly in the next line we write ““we investigate whether employing thousands of unique filters is essential for maintaining the performance of DS-CNNs.” That is why our paper is indeed novel, and as pointed by you, makes an important point. \
> These aside, we want to make a heart-to-heart statement here: Do you see the misunderstanding gap between you and us here? We know these gaps can be frustrating for you, but note that it’s hard-to-handle for us as well. Sorry if you were bothered before.
> - **“but I don't think the number of filters makes it interesting”**: That paper is not talking about the number of filters or filter redundancy, but for the point that number of filters doesn’t matter we have to strongly disagree. How come it doesn't matter if a basis filter set of master key filters are 10,000 filters or 8? Of course finding a limited set, identifying them, and reporting them is important.
> - **“This paper reduces the number of keys to 8 filters. So, what?”** We don’t know how to respond to this question. A paper not being engineering, doesn’t mean its scientific findings are “So, What?”. You can ask this question from many fundamental papers of deep learning. For example, if you were an author of >5K cited paper ”Understanding deep learning requires rethinking generalization”, and a reviewer asked you “so, what?” How would you answer? (and about the incremental part, we hope we have cleared it.)
> 2. Limited practical contribution
> - This paper contributes to our understanding of inner-working of DS-CNNs. This is clearly one of the subareas of the subject area “Deep Learning” in Neurips: “Analysis and Understanding of Deep Networks”. If this topic is not your area of interest, we understand, but it definitely is not a weakness. Also, as shown in Table 3, we reference the baseline Imagenet filters transfer which is the most common transfer learning approach.
> ---
> *To be continued on the next answer.*

---

### Comment · Area_Chair_zJZQ · 2025-08-04
**Please engage with rebuttals**

Dear reviewers,

Thank you for your work reviewing this paper. The authors have provided detailed rebuttals. Please do engage **ASAP** with the rebuttal by _updating your review_ where necessary and by _providing a final justification_, asking for further clarification where needed.

Thank you, the AC

---

### Note · Authors · 2025-08-15

We appreciate the chance for final remarks. We first briefly review our paper. Our contributions summarizes as follows:
1. This is the first paper to address DS-CNN filter redundancy and its space constraint.
2. This paper shows DS-CNN filters converge close to (aX+b) of one of 8 filters (Table 1).
3. Filters 1-4, extracted via our unsupervised methods, are novel.
4. Table 2 shows that only 8 frozen filters achieve comparable results, a first in the field.
5. Our method, using greedy search and linear approximation to identify minimal filter sets, is novel.
- **Reviewer MKLF**: We genuinely appreciate your patience for our long constructive dialogue. As discussed, we ​​performed the same experiment using the “train set” of ImageNet resulting in the same 8 filters and we’ll include that in the paper. We’ll fix notation confusion, and revise the text to better clarify our novelties.
- **Reviewer MAm3**: We thank you for improving our paper. We’ll add the proportion plots to the paper, model bank details, training/validation loss graphs, and add the “wavelet transform trick” for 3x3 filters to the appendix with more details than our discussion. We’ll also reduce the related work section and improve manuscript clarity.
- **Reviewer kyfb**: We thank you for your support and positive view in our paper. As we discussed, we will add confidence intervals, add the boxplot, and change "basis" terminology to "supporting filters".
---
As **Reviewer tTfe** did not engage in discussions, hoping they read this, we address their primary concerns again:
1. We are not using linear combinations, we use (aX+b) like (7, 7) = 2×(3, 3) + 1
2. The reviewer stated: “I hypothesize initializing randomly achieves similar results”, we conducted a new experiment showing their hypothesis doesn’t hold.
3. VAEs don’t work on our task. And they are not the “up-to-date version” of AEs. VAEs and AEs are different architectures used for different tasks.
4. We’re 100% sure the models we are experimenting with are “DS-CNN” models. We haven’t heard of any DS-CNN model whose 7x7 filter has only 14 parameters. We asked if they could please cite a model like that, but did not receive any response.

Given the reviewer's score of 2 based on the belief that even random filters perform similarly to our 8, and our experimental evidence proving this is off-base, we respectfully request score reconsideration for fair judgment, as this very major critique likely influenced the score by at least 1-2 points.

---

### Decision · Program_Chairs · 2025-09-17

**Decision:**

Accept (poster)

**Comment:**

This paper makes the compelling claim that the vast diversity of trained filters in modern Depthwise-Separable CNNs (DS-CNNs) are, in fact, predominantly simple affine transformations (ax+b) of a foundational set of just eight universal "master key" filters. The authors substantiate this by demonstrating that networks initialized with only these eight unique, frozen spatial filters can achieve competitive performance (e.g., >80% on ImageNet). This represents a significant sharpening of the recent Master Key Hypothesis to a set of only eight filters, and its experimental validation across multiple architectures and datasets. The discovery that these filters intrinsically match fundamental image processing operators (DoGs, Gaussians) provides an intriguing challenge to popular narratives around the importance of feature learning (of course, only as far as the _spatial_ filters are concerned!)

The reviewers expressed some concern regarding the relative novelty of the present work compared to previous work on the master key filter hypothesis, and the restriction to pre-LayerNorm architectures with 7x7 filters. However, given previous universality results on spatial filters in convolutional neural networks (see e.g. Guth & Ménard, 2024) I do not think that these limitations take away from the fundamental insight that this work provides. Reviewer MKLF raised a valid concern about finding the filters on the test set; during the rebuttal period, the authors reported additional experiments that assuaged this concern. During rebuttals, reviewers ​​MAm3 and kyfb, and to a lesser extent, tTfe, were convinced by the rebuttals and increased their score.

I therefore recommend acceptance, while strongly encouraging the authors to take the feedback from the reviewers into account in preparing the final version.